# Phosphorylation-deficient G-protein-biased μ-opioid receptors improve analgesia and diminish tolerance but worsen opioid side effects

A. Kliewer[1], F. Schmiedel[1], S. Sianati[2], A. Bailey[3], J.T. Bateman[4], E.S. Levitt[4], J.T. Williams[5], M.J. Christie [2] & S. Schulz [1]

Opioid analgesics are powerful pain relievers; however, over time, pain control diminishes as analgesic tolerance develops. The molecular mechanisms initiating tolerance have remained unresolved to date. We have previously shown that desensitization of the μ-opioid receptor and interaction with β-arrestins is controlled by carboxyl-terminal phosphorylation. Here we created knockin mice with a series of serine- and threonine-to-alanine mutations that render the receptor increasingly unable to recruit β-arrestins. Desensitization is inhibited in locus coeruleus neurons of mutant mice. Opioid-induced analgesia is strongly enhanced and analgesic tolerance is greatly diminished. Surprisingly, respiratory depression, constipation, and opioid withdrawal signs are unchanged or exacerbated, indicating that β-arrestin recruitment does not contribute to the severity of opioid side effects and, hence, predicting that G-protein-biased μ-agonists are still likely to elicit severe adverse effects. In conclusion, our findings identify carboxyl-terminal multisite phosphorylation as key step that drives acute μ-opioid receptor desensitization and long-term tolerance.

[1] Institute of Pharmacology and Toxicology, Jena University Hospital, Friedrich-Schiller-University, 07747 Jena, Germany. [2] Discipline of Pharmacology, School of Medical Sciences, University of Sydney, NSW 2006, Australia. [3] Institute of Medical and Biomedical Education, St George's University of London, London SW17 ORE, UK. [4] Department of Pharmacology and Therapeutics, University of Florida, Gainesville, FL 32608, USA. [5] The Vollum Institute, Oregon Health and Science University, 3181S.W. Sam Jackson Pk. Rd., Portland, OR 97239, USA. Correspondence and requests for materials should be addressed to S.S. (email: Stefan.Schulz@med.uni-jena.de)

Opioid analgesics continue to be the most effective drugs for managing severe pain, but their clinical utility is limited by life-threatening respiratory depression, and the development of analgesic tolerance and addiction after repeated administration[1]. These adverse effects underlie the alarming increase in morbidity and mortality associated with the recent escalation of abuse of prescription and potent illicit opioids. Potential mechanisms of tolerance development that drive dose escalation include agonist-induced desensitization of the μ-opioid receptor (MOP), driven by agonist-induced phosphorylation of S/T residues in the receptor carboxyl-terminal with subsequent recruitment of regulatory proteins including β-arrestins[1]. Agonist-induced desensitization and internalization of MOP occurs within seconds to minutes, whereas analgesic tolerance develops over days to weeks, because during chronic opioid exposure a number of adaptive biochemical processes occur, including upregulation of adenylyl cyclases, that may also critically support opioid tolerance[1–3]. However, the individual contributions of MOP desensitization and cellular adaptation, and the molecular mechanisms initiating analgesic tolerance remain unresolved.

MOP desensitization is thought to be initiated by the phosphorylation of S and T residues on the cytoplasmic loops and carboxyl-terminal tail following receptor activation. The carboxyl-terminal tail of MOP contains 11 S/T sites. Two-specific cassettes of MOP residues that undergo rapid agonist-induced phosphorylation, [354]TSST[357] and [370]TREHPSTANT[379], have been identified in cultured cells in vitro and in mouse brain in vivo using phosphosite-specific antibodies and quantitative mass spectrometry[4–9]. Within the [354]TSST[357] cassette, S356 and T357 are phosphorylated in an agonist-dependent manner[4,9,10], however, complete alanine mutation of the TSST motif does not affect desensitization or internalization of MOP in HEK293 cells[7,9]. Within the [370]TREHPSTANT[379] cassette, S375 in the middle of this sequence is the primary site of phosphorylation in vitro in HEK293 and AtT20 cells[6,8] as well as in mouse brain in vivo[11]. Agonists having low-efficacy for phosphorylation and β-arrestin recruitment such as morphine induce a selective phosphorylation at S375 that does not facilitate receptor internalization either in HEK293 cells[6,8] or mouse brain[11]. After overexpression of GRK2 or GRK3 in cultured cells, morphine is able to promote multisite phosphorylation and internalization of MOP[8,12]. By contrast, high-efficacy opioids such as DAMGO and fentanyl not only induce phosphorylation of S375 but also drive higher-order phosphorylation on the flanking residues T370, T376 and T379 in a hierarchical phosphorylation cascade that specifically requires G-protein-coupled receptor kinases (GRK) 2 and 3 in HEK293 cells[12] and in mouse brain[11]. This multisite phosphorylation in turn promotes both β-arrestin recruitment and robust receptor internalization in HEK293 cells[6,13]. In fact, mutation studies have revealed that mutations comprising all four S/T-residues within the [370]TREHPSTANT[379] cluster are necessary and sufficient to abolish β-arrestin recruitment and receptor internalization in HEK293 cells[6,9,13]. By contrast, the same mutations do not prevent rapid desensitization of MOP-coupling to G-protein-coupled inwardly rectifying potassium (GIRK) channels in the AtT20 cell line[9]. For complete inhibition of MOP desensitization induced by agonists having low efficacy for phosphorylation and β-arrestin recruitment, mutation of all 11 carboxyl-terminal S/T-residues to A (11S/T-A) is required[9].

To study the contribution of MOP phosphorylation for tolerance and analgesia in vivo, we created suitable mutant mouse models. Our results suggest that carboxy-terminal MOP phosphorylation is critically required for analgesic tolerance development in vivo. Phosphorylation-deficient MOP confers enhanced opioid analgesia but also increased opioid-related side effects in the absence of arrestin recruitment, suggesting that G-protein-biased opioid agonists may still possess abuse liability.

## Results

**Creation of knock-in mouse lines**. To assess the contribution of phosphorylation to MOP signalling in vivo, we created novel knock-in mice with 11S/T-A mutations (Oprm1[tm3.1Shlz], MGI:6117673, 11S/T-A). Ten of the 11S/T sites are encoded by exon 4. T394 is encoded by exon 5. Exons 4 and 5 are separated by >19 kb of intronic sequence; therefore, simultaneous mutagenesis of all 11S/T sites required knock-in of a fusion of the two exons. To produce a graded deficit in S/T phosphorylation and β-arrestin recruitment, we created another knock-in mouse line (Oprm1[tm2.1Shlz], MGI:6117668, 10S/T-A) with mutations in all 10 exon 4-encoded carboxyl-terminal S/T-residues except T394, and a third line with a single mutation of S375 to A (S375A) (Oprm1[tm1Shlz], MGI:5000465)[14] (Fig. 1a). All three transgenic lines are viable without gross phenotypic abnormalities. Autoradiographic binding assays revealed that MOP density in brains from all three knock-in lines is not significantly different from that observed in brains from C57BL/6 wild-type animals (WT) (Fig. 1b, Supplementary Fig. 1). Furthermore no differences are found in ß-arrestin-1 and GRK2 protein levels from brain lysates (Supplementary Fig. 2). In addition, MOP coupling to GIRK channels in locus coeruleus and respiratory-related Kölliker-Fuse (KF) neurons is not significantly different between WT and 10S/T-A knock-in mice (Supplementary Fig. 3).

**Lack of acute MOP desensitization**. First, we assessed the rapid desensitization of MOP coupling to GIRK channels in locus coeruleus neurons from horizontal brain sections of mice from all four genotypes. In all experiments, outward currents peak and decline during the 10 min application period of a saturating concentration of the endogenous opioid peptide [Met[5]]enkephaline (ME) (Fig. 1c). The decline from the peak values varies with genotype. In slices from WT and S375A animals, currents decrease to 50–60% of peak values, whereas the decline is much smaller in slices from 10S/T-A and 11S/T-A animals (Fig. 1c, d). Following washout of ME, effects of 1 μM ME are reduced by ~50% in slices from WT and S375A animals, but remain almost unchanged in 10S/T-A and 11S/T-A animals (Fig. 1c, d) compared to controls. Application of ME (1 μM) at various times following washout of the saturating ME concentration establishes that recovery from the smaller desensitization in slices from 10S/T-A and 11S/T-A animals is rapid, occurring within 5–10 min (Fig. 1c, d). Thus, mutation of S/T sites on the carboxyl-terminus of MOP results in a profound reduction, but not elimination, of two measures of acute desensitization. The small residual desensitization in 10S/T-A and 11S/T-A neurons resembles the heterologous desensitization that has been reported between MOP and the alpha-2-adrenoceptor upon activation of the same potassium conductance[15]. The observation that 11S/T-A and 10S/T-A mice have very similar phenotypes indicates that phosphorylation of T394 is not involved in rapid MOP desensitization.

**Enhanced opioid analgesia**. We then compared acute antinociceptive responses in S375A, 10S/T-A, 11S/T-A and WT mice in the hot-plate test. Basal pain responses do not differ between genotypes (Supplementary Fig. 4). Mice were given repeated cumulative doses of fentanyl or morphine to generate dose-response curves. Homozygous 10S/T-A and 11S/T-A animals show greater antinociception than their WT littermates (Fig. 2a). Such robust responses to fentanyl are absent in WT littermates. Interestingly, heterozygous 10S/T-A mice also exhibit significantly greater antinociception than WT mice, suggesting a

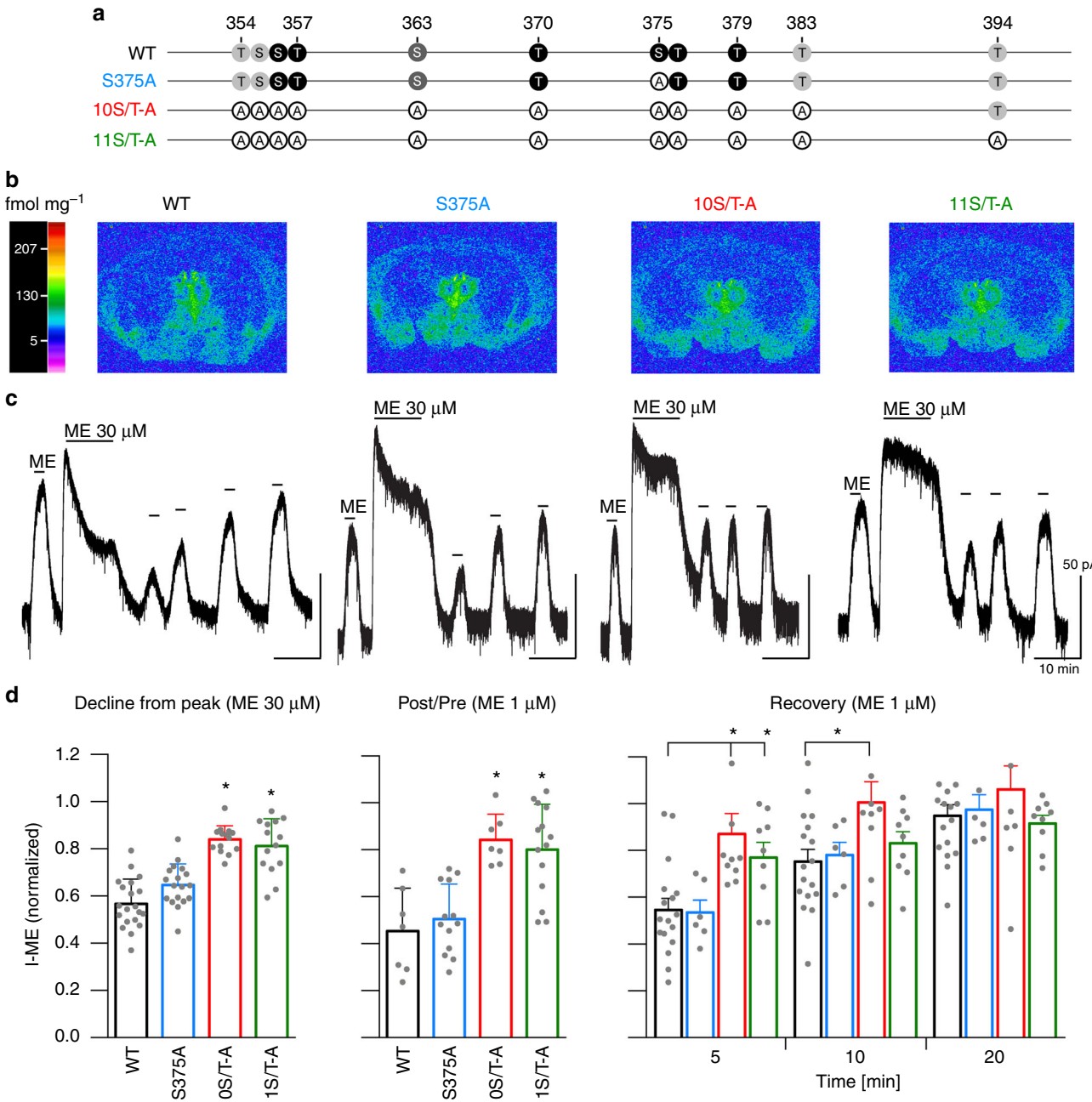

**Fig. 1** Inhibition of desensitization in phosphorylation-deficient MOP knock-in mice. **a** Schematic representation of the carboxy-terminal tail of phosphorylation-deficient MOP constructs compared to WT MOP, showing potential phosphorylation sites (black: confirmed agonist-dependent site; dark grey: confirmed constitutive site; light grey: possible site). **b** [³H]DAMGO binding to MOP as quantified in autoradiograms of coronal brain sections from WT, S375A, 10S/T-A and 11S/T-A mice at the level of the thalamus (bregma −1.46) ($n = 4$–6). The colour bar represents a pseudo-colour interpretation of black and white film images in fmol mg⁻¹ tissue. **c** Representative recordings showing acute desensitization and recovery from desensitization in locus coeruleus neurons in WT, S375A, 10S/T-A and 11S/T-A knock-in mice. [Met⁵]enkephalin (ME, 1 μM) was applied before and 5, 10 and 20 min after desensitization with a saturating concentration of ME (30 μM, 10 min). **d** Left panel: decline from peak current induced by 30 μM ME over 10 min; middle: decrease in current induced by 1 μM ME was calculated as the ratio of the current after and before desensitization (I-post/I-pre); right: recovery from desensitization measured 5, 10 and 20 min following desensitization. Data are the means ± s.e.m.; * indicates statistically significant differences compared to WT; one-way (**d**, left and middle panel) and two-way (**d**, right panel) ANOVA with Dunnett's post hoc test

dominant phenotype of the mutant receptor (Supplementary Fig. 5). In 10S/T-A and 11S/T-A mice, the maximum possible analgesic effect (30-s cutoff) is reached after administration of 0.2 mg/kg fentanyl (Fig. 2a). In contrast, in WT mice, administration of 0.3 mg/kg fentanyl is required to achieve the maximum possible effect (Fig. 2a). The 50% effective dose (ED₅₀) values determined in the hot-plate test strongly depend on the temperature, which was set at 56 °C, and cutoff for analgesia testing, which was set at 30 s. Under these conditions, the ED₅₀ value for fentanyl is twofold lower in 10S/T-A and 11S/T-A mice than in WT mice, indicating that the analgesic potency of fentanyl is increased when carboxyl-terminal phosphorylation and β-arrestin recruitment are abolished (Table 1, Fig. 2a). The duration of action of fentanyl is also prolonged (Fig. 2c,

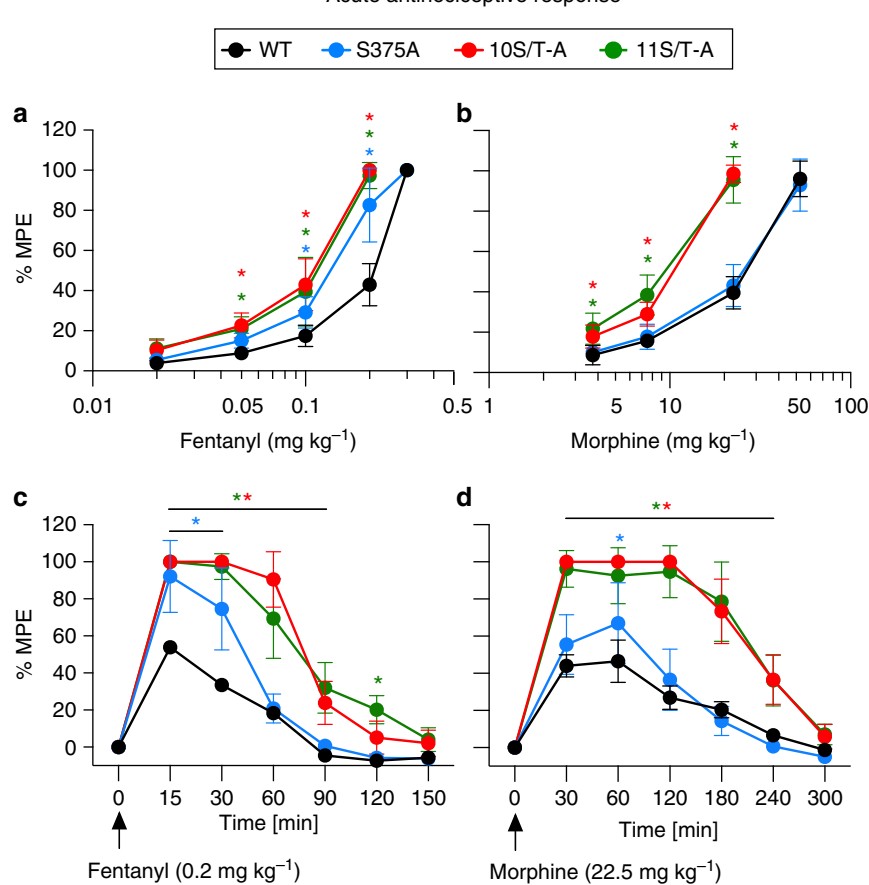

**Fig. 2** Enhanced opioid-mediated analgesia in phosphorylation-deficient MOP knock-in mice. **a–d** Acute antinociceptive response measured in the mouse hot-plate test 15 min (fentanyl) or 30 min (morphine) after drug administration. Nociceptive latencies were defined by paw withdrawal and are reported as percent maximum possible effect (% MPE) with a 30-s cutoff. **a**, **b** Cumulative dose-response curves in S375A, 10S/T-A and 11S/T-A compared to those in WT mice ($n = 9$–12) after **a**, fentanyl ($F_{\text{genotype}(3, 44)} = 62.32$, $P < 0.0001$; $F_{\text{dose}(4, 176)} = 1490$, $P < 0.0001$) or **b**, morphine ($F_{\text{genotype}(3, 39)} = 95.24$, $P < 0.0001$; $F_{\text{dose}(3, 117)} = 1363$, $P < 0.0001$). **c**, **d** Analgesic time course of acutely administered fentanyl **c**, ($F_{\text{genotype}(3, 21)} = 71.28$, $P < 0.0001$; $F_{\text{time}(6, 126)} = 463.9$, $P < 0.0001$) or morphine **d**, ($F_{\text{genotype}(3, 21)} = 66.14$, $P < 0.0001$; $F_{\text{time}(6, 126)} = 336.1$, $P < 0.0001$) was repeated after fentanyl and morphine administration, respectively, for the indicated times ($n = 6$–7). Data are the means ± s.e.m. * indicates statistically significant differences compared to WT; two-way ANOVA with Bonferroni post hoc test

**Table 1 Analgesic potency before and after chronic treatment**

| | Fentanyl (mg kg⁻¹) | | | Morphine (mg kg⁻¹) | | |
|---|---|---|---|---|---|---|
| | Day −1 | Day 7 | Fold shift | Day −1 | Day 7 | Fold shift |
| WT | 0.21 ± 0.005 | 2.35 ± 0.234$ | 11.11 ± 1.108 | 26.64 ± 1.498 | 65.11 ± 3.587$ | 2.49 ± 0.126 |
| S375A | 0.15 ± 0.008* | 0.22 ± 0.009 | 1.53 ± 0.080# | 25.09 ± 1.477 | 67.27 ± 5.690$ | 2.75 ± 0.257 |
| 10S/T-A | 0.12 ± 0.004* | 0.16 ± 0.009 | 1.35 ± 0.066# | 12.72 ± 0.222* | 22.01 ± 1.093$ | 1.74 ± 0.094# |
| 11S/T-A | 0.12 ± 0.006* | 0.21 ± 0.009 | 1.80 ± 0.121# | 11.76 ± 0.277* | 22.83 ± 1.243$ | 1.95 ± 0.108# |

Summary of pED50 values (mg kg⁻¹) for hot-plate cumulative analgesic dose-response curves before (day –1) and after (day 7) chronic treatment with osmotic pumps delivering fentanyl (2 mg/kg/day) or morphine (17 mg/kg/day) ($n = 9$–12). Data are the means ± s.e.m.
* indicates statistically significant differences in the comparison of pED50 values on day −1 to those values in WT (for fentanyl: $F_{(3, 44)} = 56.47$, $P < 0.0001$; for morphine: $F_{(3, 39)} = 52.16$, $P < 0.0001$)
$ indicates statistically significant differences in the comparison of pED50 values between days −1 and 7 (fentanyl: $F_{(7, 88)} = 86.14$, $P < 0.0001$; morphine: $F_{(7, 78)} = 75.56$, $P < 0.0001$)
# indicates statistically significant differences in the comparison of the fold shift to that in WT (fentanyl: $F_{(3, 44)} = 72.97$, $P < 0.0001$; morphine: $F_{(3, 39)} = 9.103$, $P = 0.0001$). One-way ANOVA with Bonferroni post hoc test

Supplementary Fig. 6). Furthermore 10S/T-A and 11S/T-A mice also display significantly greater and prolonged duration of antinociceptive responses to acute administration of morphine compared to WT mice (Table 1, Fig. 2b, d). Interestingly, S375A mice display greater antinociception than WT after fentanyl (but only during the first 30 min) but not after morphine (Table 1, Fig. 2). Significantly enhanced morphine analgesia in S375A mice

was only observed in time course studies at 60 min (Fig. 2d). Overall, these observations suggest that in the absence of rapid MOP desensitization, 10S/T-A and 11S/T-A receptors continue to signal for extended periods leading to increased sensitivity to both fentanyl and morphine. Thus, carboxyl-terminal multisite phosphorylation is a key determinant of MOP responsiveness in vivo.

**Exacerbated opioid side effects**. Respiratory depression and constipation are serious dose-limiting side effects of classical opioids. Previous genetic studies with β-arrestin-2 knockout mice and suggestions from some studies of biased agonists[16–20] indicate that MOP signalling through the β-arrestin pathway may contribute to these potentially lethal side effects, whereas MOP signalling via G proteins is thought to confer analgesia. However, there is no direct evidence that the severity of these adverse effects involves recruitment of the β-arrestin pathway. Because recruitment of β-arrestin proteins and MOP internalization are reduced in S375A MOP in vitro[5,8,13] and abolished in 10S/T-A and 11S/T-A mutants[6,9,13], we anticipated that total phosphorylation-deficient mice would experience significantly reduced opioid side effects. To compare the side effects across genotypes, we administered nearly equianalgesic doses of fentanyl or morphine and monitored respiration frequency, defecation and locomotor activity over time. Under these conditions, both fentanyl and morphine induce profound respiratory depression, constipation and hyperlocomotion and in all genotypes (Fig. 3, Supplementary Fig. 7a). The effects observed are almost indistinguishable between genotypes (Fig. 3, Supplementary Fig. 7a, 8). We then determined respiratory depression and constipation over a broad dose range of fentanyl or morphine and calculated $ED_{50}$ values (Supplementary Fig. 7b, c, Supplementary Table 1). Notably, we observed that 10S/T-A and 11S/T-A mice exhibit significantly greater respiratory depression than WT mice. When we plotted the $ED_{50}$ values for morphine analgesia in WT, S375A, 10S/T-A and 11S/T-A mice against their respective $ED_{50}$ values for respiratory depression and constipation we found highly significant correlations (Fig. 4, Supplementary Table 1), suggesting that both opioid effects cannot be uncoupled. KF neurons are known to contribute to opioid-induced respiratory depression[21,22]. However, MOP-mediated GIRK currents in KF neurons from 10S/T-A mice are similar to those in WT mice (Supplementary Fig. 3). Taken together, our findings suggest that the sustained G-protein signalling observed in total phosphorylation-deficient MOP knock-in mice leads to enhanced analgesia and to a proportional increase in opioid side effects that does not support a role for β-arrestin signalling in severity of respiratory depression or constipation. Conditioned place preference was also assessed after fentanyl and morphine treatment. Both WT and 11S/T-A mice clearly developed conditioned place preference (Supplementary Fig. 8a, b). However, there was no significant difference between genotypes (Supplementary Fig. 8c, d).

**Loss-of-analgesic tolerance**. In the clinical setting, analgesic tolerance usually develops with continued use of moderate levels of a drug. Given that the duration of action of fentanyl or morphine is limited to 3 or 6 h, respectively, in mice (Fig. 2c, d), chronic-treatment regimens with once- or twice-daily injections result in extended drug-free intervals. In the absence of the drug, dephosphorylation of MOP occurs within minutes[12]. To prevent MOP dephosphorylation during opioid-free intervals, we used subcutaneously implanted osmotic pumps to deliver opioids at a constant rate. This approach is a powerful means of assessing both tolerance and dependence in rodents[23]. Cumulative dose-response curves were obtained on the day before pump implantation (day −1) and 7 days after pump implantation (day 7). Tolerance was tested after a week of constant infusion of fentanyl or morphine, to eliminate potential pharmacokinetic complications. Withdrawal was precipitated with naloxone on day 8 (Fig. 5a). After chronic fentanyl treatment, WT mice show a marked rightward shift (11-fold) in potency (Table 1, Fig. 5b). Under identical conditions, S375A, 10S/T-A and 11S/T-A mice do not exhibit a significant shift in their sensitivity to fentanyl,

suggesting that tolerance to fentanyl is abrogated in these animals (Table 1, Fig. 5b). With chronic morphine treatment, mice of all four genotypes show a marked decline in response over 7 days of uninterrupted exposure (Table 1, Fig. 5c). However, the rightward potency shift observed in 10S/T-A and 11S/T-A mice is significantly less pronounced than that observed in WT and S375A mice (Table 1, Fig. 5c). There are no significant changes in MOP density levels across genotypes following chronic fentanyl or morphine treatment (Supplementary Fig. 1). These results demonstrate that tolerance to both fentanyl and morphine is profoundly blunted in 10S/T-A and 11S/T-A mice, although a modest level of tolerance to morphine remains in total phosphorylation-deficient MOP knock-in mice.

**Retained signs of physical dependence**. Prolonged exposure to opioids leads to the development of physical dependence on the drug. As tolerance is strongly diminished in phosphorylation-deficient mice, we investigated whether they would still become dependent by assessing withdrawal responses after administration of the opioid antagonist naloxone on day 8 after pump implantation. Notably, although these mouse lines differ markedly in their development of tolerance, they still display typical signs of withdrawal including jumping, wet-dog shakes and grooming. Quantitative analysis of these behaviours using a global withdrawal score[24] did not reveal significant differences between the genotypes (Fig. 5b, c, Supplementary Fig. 9). However, S375A, 10S/T-A and 11S/T-A mice display significantly greater withdrawal-associated weight loss, which is not included in the calculation of the global withdrawal score[24] (Fig. 5b, c). Although total phosphorylation-deficient mice do not become tolerant to opioids, they clearly become dependent on them.

## Discussion
By constructing mice exclusively expressing MOP receptors with carboxyl-terminal regions that progressively evade phosphorylation by GRKs[6,12] and other serine/threonine kinases (10S/T-A and 11S/T-A)[25], we have established here that carboxyl-terminal multisite phosphorylation is the proximal molecular event driving MOP desensitization and the development of long-term opioid analgesic tolerance, but not side effects or withdrawal. Although previous studies in transfected HEK293 and AtT20 cells have implicated phosphorylation by GRK2 and GRK3 and binding of the scaffolding proteins β-arrestin-1 and β-arrestin-2 in MOP desensitization[6,7,9,13,26], until now the contribution of carboxyl-terminal S/T phosphorylation to the physiological regulation of MOP has not been directly tested in vivo.

MOP desensitization in LC neurons by ME (which stimulates MOP phosphorylation and β-arrestin-2 recruitment with a similarly high efficacy to fentanyl) is not significantly affected by the S375A mutation, which had an intermediate effect on enhancing the magnitude and duration of analgesia induced by fentanyl and almost no effect on the response to morphine in vivo. It is widely established that the S375A mutation reduces overall phosphorylation of proximal residues by GRKs and strongly impairs β-arrestin-2 recruitment[6,8,11,13], without greatly affecting G-protein signalling. The increase in potency of fentanyl, but not morphine in S375A suggests that GRK/β-arrestin mechanisms contribute to limiting the acute effects of these opioids in vivo[11,14,27]. More importantly, the mutation had no differential effects on the potency of either fentanyl or morphine to induce respiratory depression or constipation, strongly suggesting that opioid side effects are mediated by a mechanism independent of β-arrestin recruitment to MOP and, therefore, pharmacological or other approaches aimed at reducing β-arrestin recruitment to MOP might not improve the safety profile

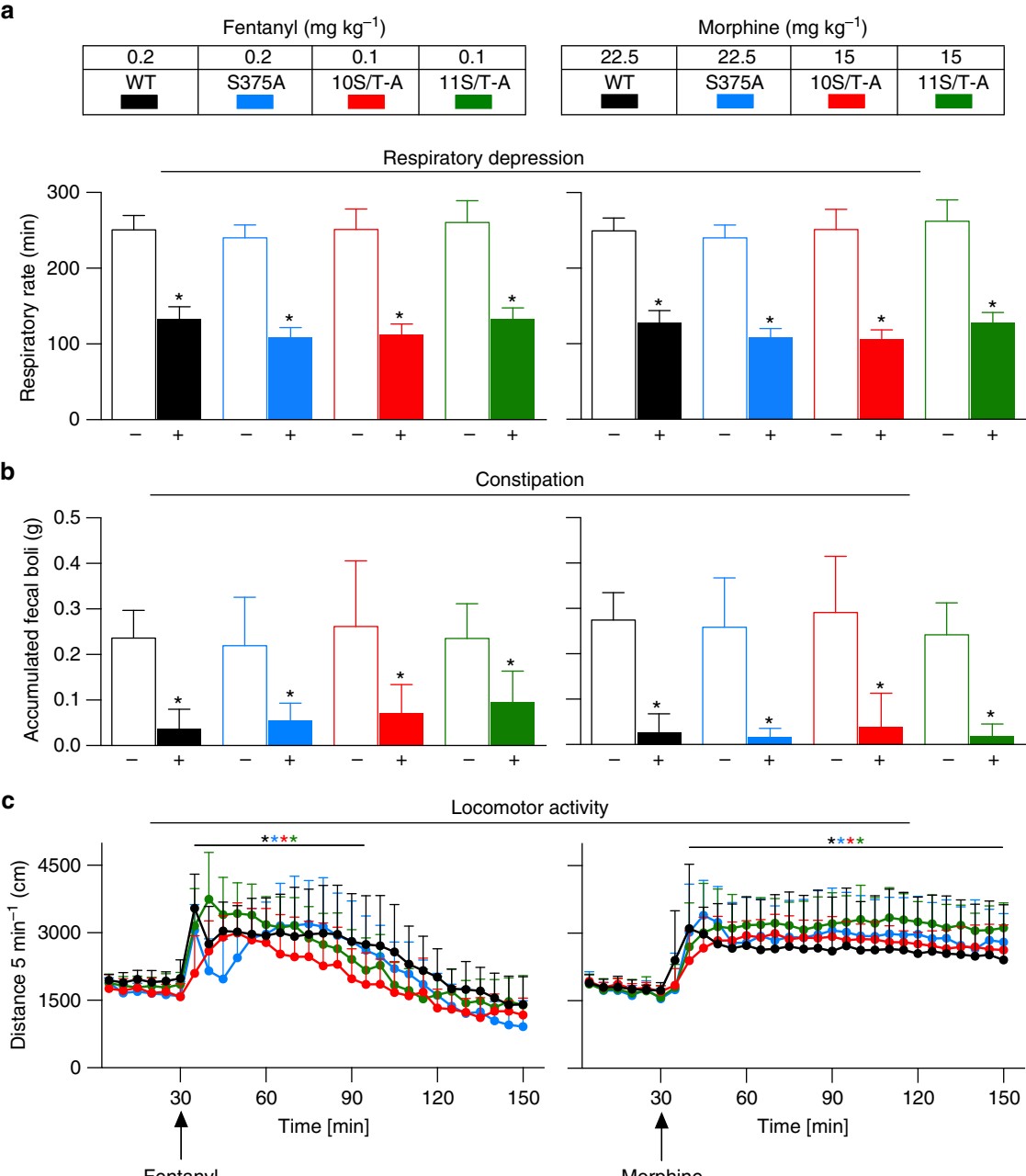

**Fig. 3** Exacerbated side effects in phosphorylation-deficient MOP knock-in mice. Mice were treated with equally effective analgesia-producing doses of fentanyl (0.1 mg kg$^{-1}$ for 10S/T-A and 11S/T-A; 0.2 mg kg$^{-1}$ for WT and S375A) or morphine (15 mg kg$^{-1}$ for 10S/T-A and 11S/T-A; 22.5 mg kg$^{-1}$ for WT and S375A). **a** Respiratory rate measured by plethysmography 15 min (fentanyl) or 30 min (morphine) after drug administration (for fentanyl: $F_{(7, 5272)}$ = 8794, $P$ < 0.0001; for morphine: $F_{(7, 5272)}$ = 10,154, $P$ < 0.0001) (n = 6). **b** Accumulated faecal boli weight in the constipation test (for fentanyl: $F_{(7, 91)}$ = 16.63, $P$ < 0.0001; for morphine: $F_{(7, 79)}$ = 26.36, $P$ < 0.0001) (n = 6–16). **c** Time course of locomotor activity in the open field test (fentanyl: $F_{time(29, 1044)}$ = 53.26, $P$ < 0.0001, $F_{genotype(3, 36)}$ = 2.684, $P$ = 0.0612; morphine: $F_{time(29, 1044)}$ = 44.28, $P$ < 0.0001, $F_{genotype(3, 36)}$ = 0.6293, $P$ = 0.06008) (n = 10). **a–c** Equally effective doses of fentanyl or morphine were administered to specific mutant strains (see Methods). Data are the means ± s.e.m.; * indicates statistically significant differences between **b**, drug (+) and vehicle (−) or **a**, **c** untreated (baseline). **a**, **b** One-way and **c** two-way ANOVA with Bonferroni post hoc test

of opioids. Complete or near complete removal of carboxyl-terminal phosphorylation sites (10S/T-A and 11S/T-A, respectively) reinforced this view, with both mutations greatly attenuating desensitization in neurons and further enhancing fentanyl and morphine potency and duration of action but without any ameliorations of the side effects. The hypothesis that β-arrestin signalling may be responsible and required for side effects such as respiratory depression is also refuted by our finding that

physiological actions of ME in KF neurons were unaffected or even enhanced in the nearly complete phosphorylation-deficient 10S/T-A mice. The lack of change in MOP density across genotypes in naive and chronic fentanyl or morphine-treated mice indicates that any phenotypic alterations observed were not due to alterations in opioid receptor levels.

The novel pharmacological concept of functional selectivity or biased agonism is based on the observation that structurally

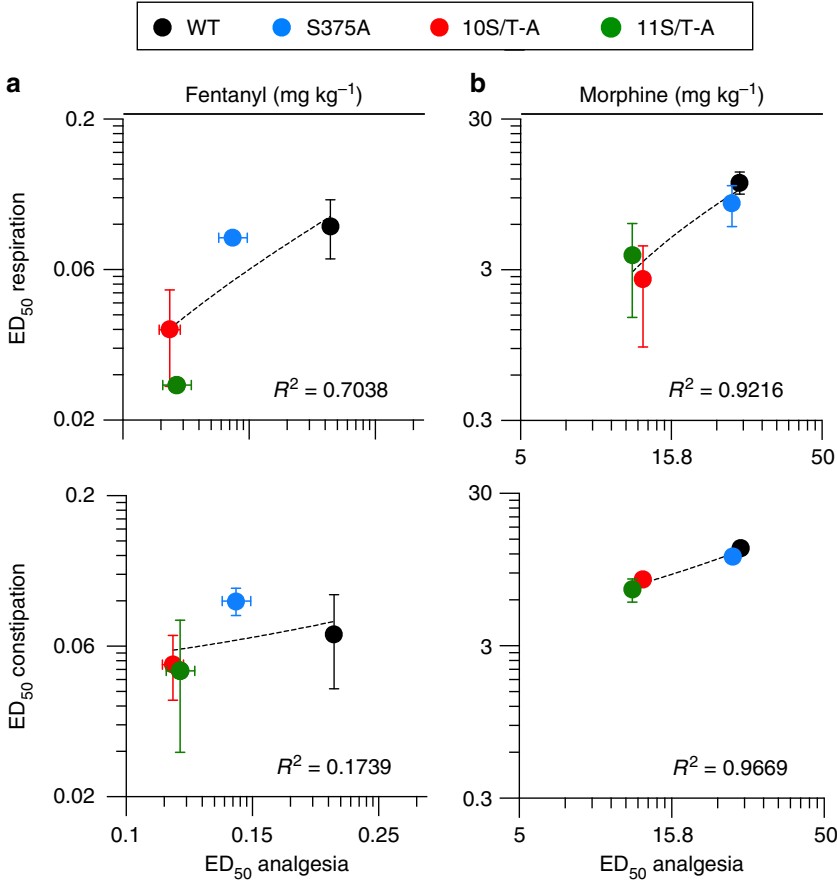

**Fig. 4** Correlation of pED$_{50}$ values for analgesia and side effects across genotypes. **a, b** Correlation analysis of pED$_{50}$ from dose-response curves for respiratory depression or constipation plotted against the pED$_{50}$ from dose-response curves for analgesia after **a**, fentanyl and **b**, morphine treatment. Data are the means ± s.e.m.; Pearson correlation for analysis of $R^2$

distinct G-protein-coupled receptor (GPCR) ligands can stabilize alternative receptor conformations upon binding, with each displaying a unique pattern of activation of intracellular signalling cascades[28,29]. Biased agonism has been proposed as a means to separate desirable from adverse drug responses downstream of a GPCR target[28,30]. Genetic studies with global β-arrestin-2 knockout mice[31] indicate that opioid side effects such as respiratory depression and constipation primarily may engage the β-arrestin-2-signalling pathway, whereas analgesia is thought to primarily engage the G-protein-signalling pathway downstream of MOP. By generating phosphorylation-deficient knock-in mice, we have created a model that allows precise genetic dissection of signalling bias at the MOP level. As 10S/T-A and 11S/T-A mutants signal via G proteins for prolonged periods without apparent desensitization but at the same time fail to recruit β-arrestin proteins, these mutants can be viewed as completely G-protein-biased MOP receptors that—contrary to prediction—do not show improved side effect profiles. This finding seems to be at odds with some observations in β-arrestin-2 knockout mice, but compensatory recruitment of β-arrestin-1 might explain the differences. We also suggest that the modest improvement in safety profile of some strongly G-protein-biased agonists may be attributable to pharmacological properties other than the loss of β-arrestin signalling. Identifying such properties may provide valuable clues for the development of safer opioids.

The profound long-term tolerance induced by fentanyl was nearly abolished in the phosphorylation-deficient mutants, but although morphine tolerance was significantly diminished, it was not completely eliminated even in mice with MOP that lacked all

possible S/T phosphorylation sites from the carboxyl-terminus. This suggests that additional mechanisms may contribute specifically to the morphine-dependent regulation of MOP. Such additional regulatory elements may include GRK5[11,12], β-arrestin-2[23] and protein kinase C (PKC)[32] or c-Jun N-terminal kinase 2 (JNK2)[33,34]. Previous studies have shown that morphine-activated MOP receptors are predominantly phosphorylated by GRK5 and preferentially recruit β-arrestin-2 but not β-arrestin-1[11–13]. While genetic inactivation of GRK5 does not affect morphine tolerance[11], global knockout of β-arrestin-2 leads to a selective attenuation of antinociceptive tolerance to morphine[23]. However, in as far as phosphorylation-deficient MOP fails to efficiently recruit β-arrestin proteins[13], β-arrestin-2 is unlikely to contribute to the residual morphine tolerance observed in 10S/T-A and 11S/T-A mice. PKC activity seems to be specifically required for morphine-dependent desensitization of MOP in LC neurons[32]. In AtT20 cells, morphine-induced desensitization was reduced by PKC inhibition in WT MOP receptors and abolished in the 11S/T-A mutant[9]. Thus, morphine may induce analgesic tolerance in 10S/T-A and 11S/T-A mice by a PKC activity-dependent mechanism that may involve phosphorylation of S/T residues in cytoplasmic loops of MOP or phosphorylation of other signalling molecules[4,35].

Superactivation of the cAMP-signalling pathway in response to chronic opioid treatment is often viewed as a cellular hallmark of opioid withdrawal[2,3,36]. Our observation that 10S/T-A and 11S/T-A mice do not become tolerant but experience all behavioural signs of withdrawal, which are mostly unchanged except for enhanced weight loss, indicates that phosphorylation-deficient

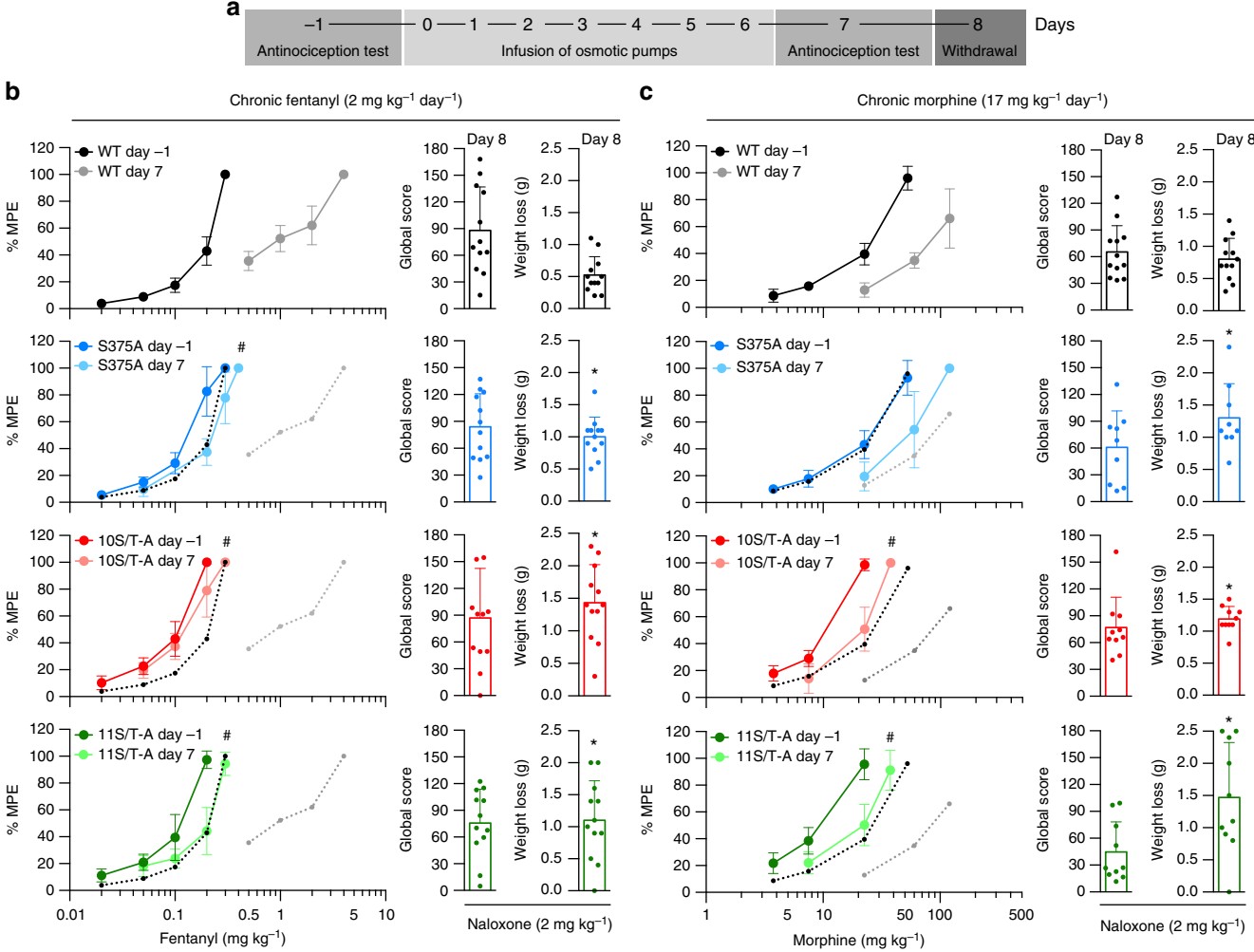

**Fig. 5** Loss of tolerance and retained signs of dependence in phosphorylation-deficient MOP knock-in mice. **a** Schematic representation of the tolerance paradigm. Acute antinociceptive response measured in the mouse hot-plate test after fentanyl (15 min) or morphine (30 min) using a cumulative dosing regimen prior (day −1) and after (day 7) to osmotic pump implantation. Withdrawal was precipitated on day 8 by naloxone injection. **b, c** Cumulative analgesic dose-response curves before (day −1) and after (day 7) chronic treatment with osmotic pumps delivering **b**, fentanyl (2 mg kg$^{-1}$ day$^{-1}$) or **c**, morphine (17 mg kg$^{-1}$ day$^{-1}$) measured in the hot-plate test ($n = 9$–12). Responses are reported as percent maximum possible effect (% MPE). The cumulative analgesic dose-response curves of WT mice before and after treatment are shown as black and grey dotted lines, respectively, in the graphs of the transgenic mice for easier comparison. The bar graphs show the global withdrawal and weight loss after 2 mg kg$^{-1}$ naloxone on day 8 ($n = 9$–12). Data are the means ± s.e.m.; $^{\#}$ indicates statistically significant differences from comparisons of the fold shift in agonist efficacy to that in WT (fentanyl: $F_{(3, 44)} = 72.97$, $P < 0.0001$; morphine: $F_{(3, 39)} = 9.103$, $P = 0.0001$), one-way ANOVA with Bonferroni post hoc test.; * indicates statistically significant differences compared to WT, unpaired, two-tailed $t$-test

mice develop the same biochemical adaptations as WT mice in the absence of tolerance. Again this observation suggests that G-protein-biased opioids may not reduce the development of opioid dependence or withdrawal after long-term treatment. The adaptive responses associated with cAMP superactivation, most notably increased expression of adenylyl cyclases[36,37], are also thought to oppose chronic opioid action, which may in turn be a source of diminished pain control[1,38]. Moreover, the RAVE (relative activity versus endocytosis) hypothesis predicts that aberrantly prolonged MOP-signalling would facilitate the development of tolerance[38]. Our findings of loss of tolerance to fentanyl and morphine but retention or enhancement of withdrawal behaviour in the phosphorylation-deficient mutants do not support the view that this type of opponent process greatly contributes to analgesic tolerance. Thus, tolerance and dependence are two dissociable phenomena and the total phosphorylation-deficient mice we have developed represent a powerful means of

dissecting the molecular pathways leading to physical dependence and addiction in a non-tolerant animal model (Fig. 6).

Finally, we indeed observed functional selectivity in these animals, in that different opioid agonists can engage different mechanisms to induce tolerance. Several recent studies have discovered novel-biased opioids with reduced side effects in animal models compared to classical opioids[18–20]. Signalling bias at opioid receptors can be envisioned at the level of $G_i$ versus $G_o$ or other G-proteins, GRK2/3 versus GRK5 or PKC, and β-arrestin-1 versus β-arrestin-2 proteins. Thus, our results indicate that the biased agonism hypothesis needs to be extended to all potential levels of bias, while simultaneously interrogating other pharmacological properties of improved opioids. Finally, our results suggest that novel chemotypes require detailed evaluation of beneficial signalling bias to allow for a precise prediction of their therapeutic window in our search for safer opioids.

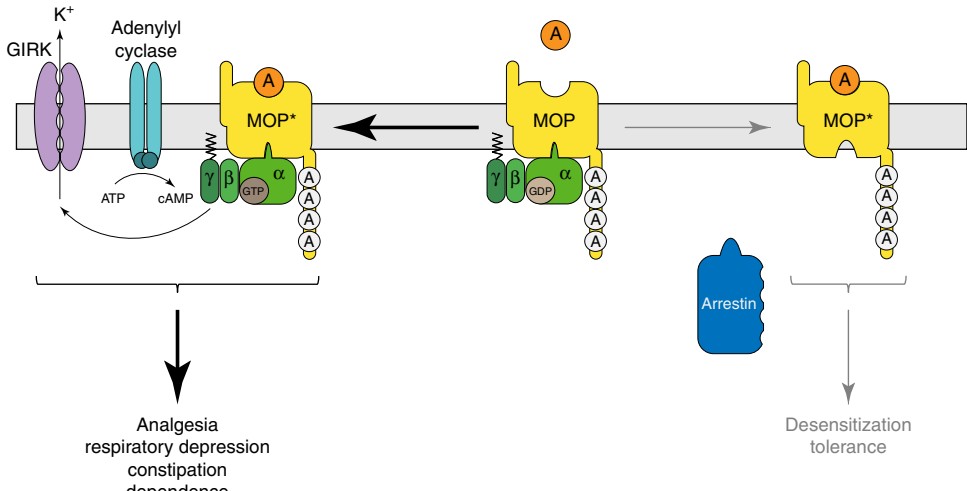

**Fig. 6** Phosphorylation-deficient, G-protein-biased MOP receptors enhance analgesia, but worsen side effects. Carboxyl-terminal multisite phosphorylation is the key step that drives acute MOP receptor desensitization and long-term tolerance. In the absence of MOP phosphorylation, arrestin recruitment is increasingly impaired yet phosphorylation-deficient MOP receptors can mediate opioid side effects such as respiratory depression, constipation and dependence

## Methods

**Animals**. Knock-in mice expressing the 10S/T-A (Oprm1[tm2.1Shlz], MGI:6117668) and 11S/T-A (Oprm1[tm3.1Shlz], MGI:6117673) mutant MOP gene were generated by Genoway (Lyon, FRA) and Ozgene (Bentley, AUS), respectively. Mice were genotyped by PCR analysis of genomic tail DNA using the following primers: 10S/T-A: 5′-TTACTATCCTCAGAGCCTTGTCTCCTTTGC-3′ and 5′-TGGGAA-TATCTTGTACCTATGACCACATTGG-3′, 11S/T-A: 5′-TTAATGTGATC-CAAGTGGGCAG-3′, 5′-TTTTGAGCAGGTTCTCCCAGTAC-3′, 5′-TTCTATCGCCTTCTTGACGAGTTC-3′ and 5′-TTAGGGCAATGGAG CAGCTTC-3′. Knock-in mice expressing the S375A MOP mutant (Oprm1[tm1Shlz], MGI:5000465) were generated and characterized as previously described[14]. All MOP mutants were backcrossed to WT control JAX[TM] C57BL/6J mice from Charles River Laboratories (DE), which were also used for the breeding of mutant strains and as controls in all experiments. WT littermates from all three MOP mutants did not differ significantly in any of the behavioural tests. For electrophysiology studies, male and female WT C57BL/6J mice (Jackson Laboratory, Sacramento, CA) were used together with all mutant strains. Animals were housed 2–5 per cage under a 12-h light–dark cycle with ad libitum access to food and water. In all behavioural experiments, we used male mice aged 8–16 weeks and weighing 25–30 g. To increase statistical power, we used 6–16 mice per genotype for each behavioural experiment group. Studies were performed in parallel such that age-matched mice received the same drug treatment at the same time. To avoid daytime effects, studies were carried out in small cohorts of mice ($n = 6$) at a time. The animal experiments conducted at Jena University Hospital were performed in accordance with the Thuringian state authorities, complied with the European Commission regulations for the care and use of laboratory animals, were in accordance with the National Institutes of Health Guide for the Care and Use of Laboratory Animals. The animal experiments conducted at University of Oregon were approved by Oregon Health and Science Institutional Animal Care and Use Committee. The animal experiments conducted at University of Sydney were performed under the guidelines of the Australian code of practice for the care and use of animals for scientific purposes (National Health and Medical Research Council, Australia, 7th Edition) and were approved by the University of Sydney Animal Ethics Committee. Our study is reported in accordance with the ARRIVE (Animal Research: Reporting of In Vivo Experiments) guidelines[39].

**Drugs and routes of administration**. All drug doses were calculated according to the active component of the salt. All drugs [morphine sulphate (3.75–120 mg kg$^{-1}$; Hameln Inc., Hameln, Germany), fentanyl citrate (0.2–4 mg kg$^{-1}$; Rotexmedica, Trittau, Germany), and naloxone hydrochloride (2 mg kg$^{-1}$; Ratiopharm, Ulm, Germany), used to precipitate withdrawal] were freshly prepared prior to use and were injected subcutaneously in lightly restrained, unanaesthetized mice at a volume of 10 μl g$^{-1}$ body weight. For chronic infusion with Alzet osmotic minipumps (1007D), fentanyl citrate salt (2 mg kg$^{-1}$ day$^{-1}$) and morphine sulphate salt pentahydrate (17 mg kg$^{-1}$ day$^{-1}$) were obtained from Sigma-Aldrich (St. Louis, MO). Drugs were diluted in phosphate-buffered saline for acute injections or dissolved in sterile water and then diluted in phosphate-buffered saline for osmotic pump delivery. For electrophysiology studies, MK-801 was obtained from HelloBio (Princeton, NJ). [Met$^5$]enkephalin (ME), bestatin and thiorphan were from Sigma-Aldrich.

**Autoradiographic binding assay**. Following decapitation, intact brains were removed, snap frozen at −20 °C in isopentane and stored at −80 °C. Adjacent sections from WT, S375A, 10S/T-A and 11S/T-A brains were cut at 300-μm intervals from fore- to hindbrain in a cryostat (Leica CM1900, UK) at −21 °C for determination of total and non-specific binding. Sections were stored at −20 °C for MOP radioligand binding as described previously[40]. Slides were pre-incubated for 30 min in 50 mM Tris-HCl, 0.9% w/v NaCl, pH 7.4, at room temperature. Slides were incubated in 50 mM Tris-HCl buffer, pH 7.4, at room temperature in the presence of 4 nM [$^3$H]DAMGO for 60 min. Non-specific binding was determined in adjacent sections in the presence of 1 μM naloxone. Incubation was terminated by rapid rinses (3 × 5 min) in ice-cold 50 mM Tris-HCl buffer, pH 7.4, at room temperature. Slides were then rapidly cool-air-dried for 2 h and dried for up to 7 days over anhydrous calcium sulphate (BDH Chemicals, Poole, UK). Adjacent total binding and non-specific sections were opposed to Kodak BioMax MR-1 film alongside $^3$H microscale standards for a period of 10 weeks for the detection of opioid receptors. Films were analysed by video-based densitometry using an MCID image analyser (Imaging Research, Canada) as previously described[41]. For each region quantified, measures were taken from both left and right hemispheres; therefore, receptor binding represents a duplicate determination for each brain region, and $n$ values refer to the number of animals analysed. The following structures were analysed by sampling 5–20 times with a box tool: cortex (8 × 8 mm), and cingulate cortex (8 × 8 mm). All other regions were analysed by freehand drawing. Brain structures were identified by reference to the mouse brain atlas[42]. Specific binding in each brain region from WT, S375A, 10S/T-A and 11S/T-A brains was compared by two-way ANOVA with Bonferroni post hoc tests.

**Brain slice preparation for electrophysiology**. Mice were anaesthetized using isoflurane (Patterson Veterinary, Devens, MA) and killed by decapitation, and brains were extracted and sectioned into 250-μm-thick horizontal slices containing the locus coeruleus (Leica VT1000S, Vashaw Scientific, Norcross, GA). Slices were cut at 32 °C in oxygenated (95% O$_2$/5% CO$_2$) artificial cerebrospinal fluid consisting of NaCl (126 mM), KCl (2.5 mM), NaH$_2$PO$_4$ (1.2 mM), MgCl$_2$ (1.2 mM), NaHCO$_3$ (21.4 mM) and D-glucose (11 mM). The cutting solution contained MK801 (0.01 mM; (5S,10R)-(+)-5-methyl-10,11-dihydro-5H-dibenzo[a,d]cyclohepten-5,10-imine; Abcam, Cambridge, UK) for blockade of NMDA glutamate receptors. Slices were transferred to a superfusion chamber maintained at 34 °C, and whole-cell recordings were made with an internal solution containing K-methanesulfonate (126 mM), NaCl (20 mM), MgCl$_2$ (1.2 mM), K-HEPES (5 mM), BAPTA (10 mM), Mg-ATP (2 mM), Na-GTP (0.25 mM) and phosphocreatine (10 mM). Cells were voltage-clamped at −55 mV and currents were recorded continuously (200/s) throughout each experiment. After establishing whole-cell recording mode, neurons were allowed to stabilize for 5–10 min prior to ME application (1 μM in the presence of 10 μM bestatin and 1 μM thiorphan to inhibit peptidase activity). Outward currents induced by ME (1 μM) ranged from 30 to 70 pA, rose to steady state within 2 min and decayed to control levels within 5 min. Following the test application of ME (1 μM), a saturating concentration of ME (30 μM) was applied for 10 min and then washed out. The decline from the peak was calculated by measuring the current at the end of a 10 min application of ME (30 μM) divided by peak current-induced shortly after (within 2 min) the application of ME (30 μM). The post/pre-measure is the amplitude of the current induced by ME (1 μM) 5 min following the washout of ME 30 μM divided by the

initial current induced by ME (1 μM). The recovery from the desensitization induced by ME (30 μM, 10 min) was measured at 5, 10 and 20 min following the washout of the high concentration of ME. The amplitude of the current induced by ME (1 μM) was measured at each time point and normalized to the amplitude of the current induced by ME prior to the application of the high concentration of ME. Significant differences between groups were determined by unpaired two-tailed $t$-tests or by two-way ANOVA with Bonferroni or Dunnett's post hoc tests. Values are given as the means ± s.e.m. and $n$ = number of cells.

**Osmotic pump implantation**. A single 1007D Alzet osmotic minipump (Charles River Laboratories, DE) was subcutaneously implanted on the left limb of each mouse under light isoflurane anaesthesia and meloxicam analgesia 7 days before behavioural experiments. A small incision was made in the skin on the mouse's left flank with a scalpel, a small pocket was formed just beneath the skin, and the minipump was inserted. The incision was closed using 7-mm wound clips (Charles River Laboratories).

**Behaviour studies**. For consistency, one experimenter and a dedicated assistant performed all in vivo drug administrations and behavioural testing. All testing was conducted between 7 a.m. and 4 p.m. in an isolated, temperature- and light-controlled room. Mice were acclimated for at least 2 weeks before testing. Only the experimenter and assistant had free access to the room and entered the room 30 min before commencement of testing to eliminate potential olfactory-induced changes in nociception. Animals were assigned to groups randomly before testing. Mice were excluded from the study if they displayed any bodily injuries from aggressions with cage mates. The experimenter was blinded to treatment and/or genotype throughout the course of behavioural testing. All drugs were given to the experimenter in coded vials and decoded after completion of the experiment.

Hot-plate test. Opiate effects on paw withdrawal latencies were assessed as the time to response (licking or flicking fore or hind paw(s)) after placement on a hot-plate maintained at 56 °C (Ugo Basile SRL, IT)[11,14]. To avoid tissue damage, we used a 30-s cutoff. The hot-plate test was carried out 15 min (fentanyl) or 30 min (morphine) after drug administration and expressed as percent maximum possible effect (% MPE), calculated as follows: $100 \times$ [(drug response latency–basal response latency)/(30 s–basal response latency)]. For time courses antinociception was determined at various time points after treatment.

Acute analgesia and tolerance paradigms. For acute analgesia testing in the hot-plate test, dose-response curves were generated after repeated subcutaneous administration of cumulative doses of fentanyl or morphine. Mice were injected at 15 min intervals with 0.02, 0.03, 0.05, 0.1 and 0.1 mg kg$^{-1}$ fentanyl to yield final cumulative doses of 0.02, 0.5, 0.1, 0.2 and 0.3 mg kg$^{-1}$ fentanyl, and latencies were measured 15 min after drug administration, immediately followed by additional drug except for the last dose. For morphine dose-responses, mice were injected at 30 min intervals with 3.75, 3.75, 15 and 30 mg kg$^{-1}$ morphine resulting in final cumulative doses of 3.75, 7.5, 22.5 and 52.5 mg kg$^{-1}$ morphine. Hot-plate latencies were always measured 30 min after morphine administration, immediately followed by the next dose. Dose-response curves generated using repeated cumulative dosing regimen are shown in Fig. 2a, b. These curves were used to calculate ED$_{50}$ values for analgesia shown in Table 1 and Fig. 4. To induce opioid tolerance, mice were implanted one day later with Alzet osmotic minipumps containing the same drug that was used in the cumulative dosing. Osmotic minipumps delivered total daily doses of 2 mg kg$^{-1}$ fentanyl or 17 mg kg$^{-1}$ morphine at a rate of 0.5 μl/h. Thus, in the tolerance paradigm dose-response curves from acute analgesia testing are depicted as day −1 in Fig. 5 and served as reference for the development of tolerance. On day 7, mice were again treated using a repeated cumulative dosing regimen with fentanyl (0.05, 0.05, 0.1, 0.1 mg kg$^{-1}$ for 10S/T-A and 11S/T-A and 0.05, 0.15, 0.1, 0.1 mg kg$^{-1}$ for S375A and 0.5, 0.5, 1, 2 mg kg$^{-1}$ for WT) or morphine (7.5, 15, 15 mg kg$^{-1}$ for 10S/T-A and 11S/T-A and 22.5, 37.5, 60 mg kg$^{-1}$ for S375A and WT), and hot-plate response latencies were assessed at the same time points as on day −1. ED$_{50}$ values calculated from day −1 and day 7 dose-response curves were used to calculate the fold rightward shift given in Table 1 as a measure of tolerance."

Physical dependence studies. After induction of chronic tolerance, withdrawal was precipitated on day 8 by 2 mg kg$^{-1}$ naloxone HCl injection. Mice were individually placed in small Plexiglas boxes (26.5 cm × 20.5 cm × 28 cm) lined with filter paper, and observed and scored for 20 min for manifestation of different withdrawal signs, including jumping, grooming and wet-dog shakes. Weight loss was determined by subtracting the measured body weight after withdrawal precipitation from that prior to withdrawal[23]. A global withdrawal score, excluding weight loss, was calculated as previously described[23,24]. Withdrawal signs were weighted as follows: jumps, 0.8; wet-dog shakes, 1, and grooming 0.35. The sum of all weighted signs produced a global withdrawal score for each mouse[23,24,42].

Mouse plethysmography. Respiratory rates were recorded with a nose-out plethysmography system (Hugo Sachs Elektronik–Harvard Apparatus GmbH, DE). Individual unanaesthetized mice were placed in a restrainer with their nose exposed through a close-fitting hole in the membrane. A pneumotachograph was connected to the chamber equipped with a differential low-pressure transducer (transducer DLP2.5, Hugo Sachs Elektronik–Harvard Apparatus GmbH)[43]. Volume changes were calibrated by injecting known amounts of air into the chamber. Analogue pressure signals were digitized for later analysis[44]. Mice were acclimated to the test chamber, and baseline respiratory parameters were recorded

for 30 min. The 30-min test was then repeated 15 min (fentanyl) or 30 min (morphine) after injection of equally effective doses of fentanyl (0.1 mg kg$^{-1}$ for 10S/T-A and 11S/T-A; 0.2 mg kg$^{-1}$ for WT and S375A) or morphine (15 mg kg$^{-1}$ for 10S/T-A and 11S/T-A; 22.5 mg kg$^{-1}$ for WT and S375A) or as indicated. Data were analysed in Pulmodyn® W Software (HSE – Harvard Apparatus GmbH, DE). For dose-response curves, percent maximum possible effect (% MPE) were calculated as follows: $100 \times$ [(drug response–average baseline)/(maximal response cutoff–average baseline)]. The maximum response cutoff for respiratory rate was set at 60 breaths per min.

Accumulated faecal boli quantification. Mice were subcutaneously injected with vehicle or equally effective doses of fentanyl (0.1 mg kg$^{-1}$ for 10S/T-A and 11S/T-A, and 0.2 mg kg$^{-1}$ for WT and S375A) or morphine (15 mg kg$^{-1}$ for 10S/T-A and 11S/T-A, and 22.5 mg kg$^{-1}$ for WT and S375A) or as indicated and individually placed into small Plexiglas boxes (26.5 cm × 20.5 cm × 14 cm) lined with filter paper. Faecal boli were collected and weighed every hour for 3 (fentanyl) or 5 (morphine) hours[31]. For dose-response curves, percent maximum possible effect (% MPE) were calculated as follows: $100 \times$ [(drug response–average baseline)/(maximal response cutoff–average baseline)]. The maximum response cutoff was set at 0 g faecal boli.

Open field locomotion test. The locomotor activity of mice was monitored individually in open-field boxes (50 cm × 50 cm × 50 cm) using a TSE Videomot system (TSE Systems, Bad Homburg, DE). Mice were habituated for 30 min, removed, injected with equally effective doses of fentanyl (0.1 mg kg$^{-1}$ for 10S/T-A and 11S/T-A and 0.2 mg kg$^{-1}$ for WT and S375A) or morphine (15 mg kg$^{-1}$ for 10S/T-A and 11S/T-A and 22.5 mg kg$^{-1}$ for S375A and WT) and immediately placed back into the box for 120 min of monitoring[31].

**Data and statistical analysis**. All experiments were randomized, performed by a blinded researcher, and then unblinded before statistical analysis. The results for each experiment were expressed as the means ± s.e.m. For ED$_{50}$ values of dose-response curves, the best-fit line was generated following nonlinear regression analysis based on the % MPE, as described in each behavioural method section above. Fold shift in effect was calculated by dividing values for day −1 by those for day 7. Area under the curve was calculated for each animal with the baseline defined as zero. Normal distribution of the data was verified before performing parametric statistical analysis. Wherever appropriate, data were analysed using one-way or two-way ANOVA, followed by Bonferroni's or Dunnett's post hoc tests or unpaired, two-tailed $t$-test with significance set at $P < 0.05$. Pearson correlation was used for analysis of $R^2$ with significance set at $P < 0.05$. All calculations were performed using GraphPad Prism 6 software (GraphPad Software, Inc., San Diego, CA).

**Reporting summary**. Further information on experimental design is available in the Nature Research Reporting Summary linked to this article.

## Data availability
The authors declare that all data supporting the findings of this study are available within the paper and its supplementary information files. The source data underlying all figures are provided as a Source Data file. The data that support the findings of this study are available from the authors upon reasonable request.

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

## Acknowledgements

We thank Helga Bechmann, Elke Miess, Pooja Dasgupta, Sebastian Fritzwanker and Heike Stadtler for excellent technical assistance, Ralf Stumm for discussion and Rainer Reinscheid for critical reading of the manuscript. This work was supported by the Deutsche Forschungsgemeinschaft grants SFB/TR166-TPC5 and SCHU924/10-3 to S.S., NIH DA08163 to J.T.W., National Health and Medical Research Council of Australia (APP1072113 and 1045964) to M.J.C., Horizon 2020 EU funding SmokeFreeBrain 681120 to A.B. and NIH DA038069 to E.S.L.

## Author contributions

S.S. initiated the project and designed all behavioural pharmacology experiments with A. K. A.K. performed all in vivo studies. F.S. performed mouse plethysmography, accumulated faecal boli studies and CPP. S.Si., M.J.C. and J.T.W. performed electrophysiology on the locus coeruleus and E.S.L. and J.T.B. performed the experiments on the Kölliker-Fuse nuclei. A.B. performed the autoradiographic binding assay. The manuscript was written by S.S., A.K. and M.J.C. with editing and suggestions from A.B.

## Additional information

**Competing interests:** The authors declare no competing interests.

