## [Peer Review File · Nature Communications]

Reviewers' comments:

Reviewer #1 (Remarks to the Author):

In this interesting and intriguing manuscript, the authors aim to provide necessary insight into the mechanisms underlying mu opioid receptor (MOP) desensitization that drives opioid side effects such as analgesic tolerance, respiratory depression, GI motility and dependence.

MOP desensitization is thought to be mediated via phosphorylation of S and T residues at the cytoplasmic loops and carboxy-terminal tail and subsequent recruitment of regulatory proteins including B-arrestins following agonist-evoked activation. Work conducted in the laboratory of Dr Schulz using phospho-specific antibodies in combination with mass spectrometry has identified the specific location and sites for agonist-evoked MOP phosphorylation. Based on this, the authors have designed 3 different knockin mouse lines with a series of carboxy-terminal S/T to alanine mutations at the MOP that are not able to recruit regulatory arrestin proteins following agonist binding. In this study, the authors examine whether analgesic properties of opioids, tolerance and other adverse effects are affected in these knockin mice.

Interestingly, they observe that mice with phosphorylation deficient MOP exhibit enhanced analgesia to morphine and the short-acting opioid fentanyl. In addition, a reduction or elimination of analgesic tolerance following chronic opioid administration is also observed. However, other chronic opioid-associated side effects such as GI motility, opioid withdrawal signs or respiratory depression are not prevented or they are enhanced which seems at odds with previous literature on biased agonist to prevent these side effects.

Although the data presented are solid and the knockin mice seem to be an interesting model to examine the role of MOP phosphorylation on opioid-evoked responses further studies and an expanded discussion needs to be provided in order to reconcile the current studies with prior studies highlighting the role of B-arrestin regulatory proteins on the undesirable effects of chronic opioid administration.

As discussed in the introduction, opioid tolerance may lead to drug escalation and ultimately misuse and addiction. But this study doesn't examine whether opioid reward may also be affected by phosphorylation deficient MOP. Therefore, additional studies are needed to examine this using appropriate animal models of dependence and opioid seeking behavior.

In addition, studies under conditions of pain need to be conducted as well to examine whether the observed enhanced analgesic properties of opioids are also observed in the setting of pain.

Finally, there are some procedures that need to be clarified and better justified. For example, opioid tolerance is induced following a 2-step procedure. First animals are injected with increasing concentrations of the opioid and then an osmotic minipump is inserted. The authors provide justification of the use of the minipump in order to maintain continuous opioid delivery but they don't provide justification for the intermittent administration of the drug prior to the insertion of the osmotic pump.

Reviewer #2 (Remarks to the Author):

Please note: I wish my report/comments to be anonymous.

This manuscript describes the effects of opioid agonists in knock-in mice where the C-terminus of the mu receptor (MOP) has had the serine/threonine residues mutated to alanines, and so cannot be phosphorylated in the C-terminus. The inference is that these mutant receptors cannot recruit β arrestins, and this has been shown in previous work in cell lines. The results of the present study clearly show that under these conditions, and compared to wild type mice, MOP in the mutant mice

are less able to undergo agonist-induced desensitization over a short time frame of a few minutes. In addition the agonists fentanyl and morphine are more potent and longer lasting when tested for acute analgesia, and importantly other effects of these agonists (respiratory depression, constipation and locomotor activity) are not reduced in the absence of MOP C-terminus phosphorylation, and presumably β arrestin recruitment. Furthermore whilst the long term analgesic tolerance to fentanyl and to some extent morphine, is also removed or reduced, the withdrawal effects when naloxone is administered are not significantly altered.

In general the quality of the experimental work is high and clear conclusions can be reached.

The findings are of importance for a number of reasons

- The opioid receptor and the associated drugs such as morphine and fentanyl are of very great importance in both medicine (analgesia etc) and society (drug abuse).
- The molecular mechanism of opioid tolerance and its relationship to receptor phosphorylation is of great interest in the opioid and more general GPCR field
- For many years the opioid field has been influenced by previous findings implying that G proteins mediate MOP agonist analgesia, whilst β arrestins mediate the adverse effects such as respiratory depression. This idea has really dominated the opioid field for well over a decade, but the present results clearly overturn that idea. The work is thus very important and will have significant influence on the future direction of the opioid field.

Other comments

1. Should not fentanyl and morphine have been used for the brain slice recording? Why was Met Enk used instead? At least some results with fentanyl and morphine should be included to show that these behave similarly to MetEnk.
2. Presumably it has been done but are brain GRK and β arrestin levels the same in the different mouse lines?
3. It would have been useful to have some further brain biochemistry results to support the findings. For example, in the different mouse lines what happens to brain cAMP levels with naloxone challenge after chronic opioid? The behavioural results predict that the effects will be the same in all the mouse lines. Also it would be really good if it could be shown that in neurones from the mouse brains the mutant receptors do not recruit β arrestins. Is it possible that limited recruitment of arrestins could occur due to receptor activation alone, and independent of GRK-phosphorylation?
4. Do the mutant receptors still recruit GRKs? If so could this mediate effects independent of MOP phosphorylation? Could the mutant C-terminus of MOP interact with novel proteins?
5. I am not exactly sure what Figure 4 is meant to show, or what we are meant to conclude from it. The R2 for morphine are significant whilst those of fentanyl are not, but what are we meant to take from this? These sort of correlations with 4 points don't seem that useful.
6. In the title the work "exacerbation" does not seem to be strongly supported by the results. It may be better to alter this to "maintenance of.." or something similar. Figure 3 and Ext data Figure 6 show that fentanyl and morphine become more potent, but the extent of say the respiratory depression is not increased. If the authors want to retain "exacerbation" then they should explain clearly why this description is justified in the discussion.
7. Figure 3 – it would be good to see a time course of the respiratory depression in the different mouse lines and for the two agonists – are these the same? These could be included in the Ext data. I am also not clear why vehicle treatment is not used for comparison in parts a and c? In part c, the time course of the S375A mutant is strange. Any comment? Also check the X-axis label in c – is it correct?
8. P12 lines 4-10. Be specific in the studies being referred to here – which is the work of Bohn and colleagues, which has dominated the field for years.
9. P13 lines 16-20 – I was amused to see the RAVE idea being resurrected here – it is fine to leave as is, but it has been largely/totally discredited really
10. Figure 5 – from visual inspection of the data, it seems to me that there is still significant tolerance to morphine in the mutant mice, which doesn't seem that different from the wild type

when the shift is considered. Any comment on this?

11. P9 lines 7-8 – Some data should be included to show that the chronic drug treatment did not affect receptor levels. There is conflicting data on this in the literature.
12. Figure 5 – The global score for morphine in 11S/T-A is somewhat less than for the others – any reasons for this? Also it is not clear what we are meant to conclude from the weight loss data in general, e.g. the difference between morphine in 11S/T-A and wild type?
13. P2 line 18 – better to say at end of line “but that severe adverse effects are retained”?
14. P3 line 12-14. Sentence grammatically incorrect.
15. P6 line 4 – mention type of test used at beginning.
16. P6 line 8 – are stats included for the heterozygous mice? Can’t see in the Ext data.
17. P6 and elsewhere. The EC50 values depend on the cut-off for analgesia testing. Whilst the data presented is fine, this consideration should be mentioned somewhere. A statement such as “maximal possible analgesic effect” on P6 line 11 can be a little misleading.
18. P7 lines 5 and 8 – not sure constipation is normally regarded as a lethal effect with opioids?
19. P7 lines 21-22 – better to say that at low opioid drug concentrations they produced sig greater resp depression and constipation?
20. P8 line 8 – again not sure it is right to refer to a “proportional increase in opioid side effects”
21. Fig5 – how were chronic doses decided upon/calculated?
22. P9 line 2 – “a significant shift in their” better
23. P30 line 12. N=4 here but the data in panel d shows many more experiments (dots). Not clear why this is.
24. P31 line 13 – “equally effective doses for analgesia of..”
25. P32 line 14 – b,c rather than a,c I think
26. P32 line 20 and Fig 5 – does naloxone produce any scores if given to untreated mice, or for that matter, does vehicle produce any effects in treated mice?

Reviewer #3 (Remarks to the Author):

Review NCOMMS-18-15778-T

Kliwer et al. “Enhanced opioid analgesia and loss of tolerance but exacerbated side effects in mice expressing G-protein biased, phosphorylation-deficient μ -receptor”

General comments:

This is a very interesting paper that examines mechanistic aspects of opioid receptor signaling with important implications for opioid analgesia, side effects and drug discovery. The approach is highly innovative; the topic is current. The authors are highly regarded experts in this field.

The authors show that mutating 10 (S/T) phosphorylation sites at the C-terminus of the mu-opioid receptor (MOR) prolongs and enhances analgesia, reduces opioid tolerance but enhances adverse side effects. These results might significantly influence current trends in the design of opioid agonists (e.g. G-protein biased ligands). However, some important data are missing. Whereas desensitization is clearly abolished/diminished in S/T- mutants, no direct evidence of phosphorylation-deficiency, internalization or beta-arrestin recruitment is shown here. If the authors have such evidence, this must be included in the MS. The discrepant results about the role of S375 in this paper compared to previous publications (Reference No. 14) need to be discussed, as they raise concerns about reproducibility of the data. There are also some important issues with data presentation (were some of the data presented multiple times without stating so?), study design and statistical analysis.

Specific comments:

The abstract (p.2 line 9) (and many text passages throughout the MS) contains statements about the lack of arrestin recruitment in these mutants. However, no data on arrestin recruitment (or receptor internalization/phosphorylation) are shown here. Such data need to be included in the present MS. Instead, the authors refer to a paper in press which is not available to date.

In the introduction the authors make several statements concerning the relevance of amino acid residues for the phosphorylation/desensitization/internalization of opioid receptors. Some of these statements are too general because they do not specify the cells/tissues/animals that were used to generate these data in previous studies (e.g. cell lines, brain regions, rats, mice, humans etc.). This important information needs to be added. Similarly, the terms "low-efficacy agonist" (morphine) and "high-efficacy agonist" (fentanyl) (p. 4, line 3-4) are too general. Which type of efficacy (analgesia, side effects, animal species, cell line, signaling pathway) do the authors refer to? Do they really mean "efficacy" or, rather, "potency"?

Fig 2, Fig 5: Are these the same data/animals that are shown in Fig. 4? If so, this should be stated. The cumulative dose regimen needs to be described in more detail: the individual doses need to be stated; are the same animals injected multiple times? Is this procedure stressful for the animals?

On p. 6, line 18-20 it is stated: "Interestingly, S375A mice display greater antinociception than WT after fentanyl (but only during the first 30 min) not but after morphine (Table 1, Fig. 2)": The lack of enhancement of morphine analgesia is in contrast to results previously published in Grecksch et al. 2011 (Fig. 4 in that paper). The authors should address this reproducibility issue. A direct comparison of the doses/effects is also made difficult by the confusion about the employed doses in the present paper.

p. 7 Line 10-12: Whereas MOR internalization has been investigated in earlier papers, direct effects on arrestin recruitment are not shown in the cited papers (No. 5-9). The results presented in paper 13 are not available to date (paper in press?) making the appropriate assessment of the present MS impossible. Similar to the introduction, whenever data are compared between the present and previous manuscripts, the assays and tissues (cell lines, species, brain regions etc.) used in the different publications need to be clearly stated in order to enable the reader to adequately assess comparability.

p. 7 line 14: "...administered equi-analgesic doses of fentanyl or morphine ...". Are these doses really equally effective in analgesia? In Fig. 2 the S375A mutant seems different at 0.2 mg/kg compared to WT (same dose), and to 10S/T-A and 11S/T-A at 0.1 mg/kg.

p. 7 line 21-22: "... that 10S/T-A or 11S/T-A mice experienced significantly greater respiratory depression and constipation than WT mice." It seems that this sentence is not accurate for fentanyl, since the ED50s for constipation do not significantly differ from WT. Also, despite the clear differences in the ED50s (Ext. Table 1) for respiratory depression and constipation, the dose response graphs in Ext. fig 6 do not clearly reflect such differences. Perhaps the authors should opt for a different graphical representation which highlights better the ED50s shifts and accurately describe the way they calculated the ED50s from such dose response experiments.

The following statements are inaccurate/misleading:

p. 8 line 1-2 (and Fig 4): the "highly significant correlation" seems to be the case for morphine only and not for fentanyl. It would be interesting to discuss this discrepancy.

p. 9 line 5-6 (Fig. 5): "However, the rightward 5 potency shift observed in 11S/T-A and 10S/T-A mice was significantly less pronounced than that observed in WT and.." A statistical test demonstrating this significance is missing.

p. 9 line 14 (Fig 5): "phosphorylation-deficient mice do not become tolerant," This statement should be rephrased, as the authors claim elsewhere "a modest level of tolerance to morphine remains.."

p. 10 line 17-18: "... which stimulates MOP phosphorylation and β arrestin-2 recruitment with a similarly high efficacy as fentanyl". References to this statement are missing.

p. 11 line 1 and Fig. 2b: The authors should discuss the discrepancy between this result and those presented in Grecksch et al. 2011

p. 12 line 2-4: "... at the same time fail to recruit β -arrestin proteins, these mutants can be viewed as completely G protein-biased MOP receptors that do not show improved side effect profiles". These statements are not supported by the data included in the present MS. If the authors have direct evidence of lack of arrestin/internalization/phosphorylation and G-protein bias in the cells/tissues from these knockin animals, such evidence should be included in the MS or the statement would need to be revised.

p. 13 line 11-12: "... enhanced weight loss, indicates that phosphorylation-deficient mice develop the same biochemical adaptations as WT mice in the absence of tolerance". Without cAMP measurements, this statement is speculation, considering the very general/nonspecific observation of weight loss. The enhancement of weight loss (especially in the 11S/T-A mutants) should be more carefully investigated, particularly since the treatment/testing paradigm applied here seems to be quite challenging for the animals. Moreover, the effect seems to be very small especially in the case of the 10S/T-A mutants. Therefore, the employed statistical tests should be double-checked.

Methods:

p. 19, line 1: A description of how the data in Fig. 1d were generated (formulas of the ratios, normalization, which timepoints e.g. in the case of post/pre were compared etc.) would be helpful.

p. 19, line 5-10: Were the animals used for the minipump implantation the same that were tested in Fig. 2 and 5? Were the chronic opioid administration experiments blinded?

p. 20, line 4-7: To improve the readability of the paper and the understanding of the results, the doses/injection regimen employed for these tests should be mentioned here. If these animals were then subjected to minipump implantation and to subsequent testing, this should be stated.

p. 20, line 9-11: there must be several typos in the indication of the doses employed.

From Fig. 2b it seems that also 50 mg/kg was employed

p. 22, line 21: Normal distribution of the data must be verified before performing parametric statistical analysis.

p. 24, line 4-6. This paper is not available to date (in press?). No data on arrestin recruitment are shown in the present MS. However, such data are essential for adequate evaluation of the present results, particularly when it comes to the patterns of recruitment of beta-arrestin by the activated phosphorylation-deficient MOP.

Fig. 1c: which time point is represented by the last peak at the right side?

October 10, 2018

Fiona Carr, PhD
Associate Editor
Nature Communications

Revision of manuscript: NCOMMS-18-15778-T

Dear Fiona,

We are grateful to the Reviewers and the Editorial Board for critically reviewing our manuscript entitled “*Enhanced opioid analgesia and loss of tolerance but exacerbation of side effects in mice expressing phosphorylation-deficient, G protein-biased μ -receptors*”. We have now completed the revision of our manuscript, taking all of the Reviewers' comments and criticisms into account. They prompted revision of the text and figures and additional experiments, which has strengthened our manuscript. Please see the detailed responses below (blue text).

Reviewer 1 comments

In this interesting and intriguing manuscript, the authors aim to provide necessary insight into the mechanisms underlying mu opioid receptor (MOP) desensitization that drives opioid side effects such as analgesic tolerance, respiratory depression, GI motility and dependence. MOP desensitization is thought to be mediated via phosphorylation of S and T residues at the cytoplasmic loops and carboxy-terminal tail and subsequent recruitment of regulatory proteins including B-arrestins following agonist-evoked activation. Work conducted in the laboratory of Dr Schulz using phospho-specific antibodies in combination with mass spectrometry has identified the specific location and sites for agonist-evoked MOP phosphorylation. Based on this, the authors have designed 3 different knockin mouse lines with a series of carboxy-terminal S/T to alanine mutations at the MOP that are not able to recruit regulatory arrestin proteins following agonist binding. In this study, the authors examine whether analgesic properties of opioids, tolerance and other adverse effects are affected in these knockin mice. Interestingly, they observe that mice with phosphorylation

deficient MOP exhibit enhanced analgesia to morphine and the short-acting opioid fentanyl. In addition, a reduction or elimination of analgesic tolerance following chronic opioid administration is also observed. However, other chronic opioid-associated side effects such as GI motility, opioid withdrawal signs or respiratory depression are not prevented or they are enhanced which seems at odds with previous literature on biased agonist to prevent these side effects. Although the data presented are solid and the knockin mice seem to be an interesting model to examine the role of MOP phosphorylation on opioid-evoked responses further studies and an expanded discussion needs to be provided in order to reconcile the current studies with prior studies highlighting the role of B-arrestin regulatory proteins on the undesirable effects of chronic opioid administration.

1. As discussed in the introduction, opioid tolerance may lead to drug escalation and ultimately misuse and addiction. But this study doesn't examine whether opioid reward may also be affected by phosphorylation deficient MOP. Therefore, additional studies are needed to examine this using appropriate animal models of dependence and opioid seeking behaviour.

We agree with Reviewer 1. To address this important question we have performed conditioned place preference (CPP) assays with 11S/T-A and WT mice. We were able to show that these mice develop CPP to morphine and fentanyl in a manner similar to wild-type mice, suggesting that opioid seeking behaviour is retained in phosphorylation-deficient mice. These new results were added as Supplementary Fig. 8 and described in the method section. We have added a comment on page 8 line 6.

2. In addition, studies under conditions of pain need to be conducted as well to examine whether the observed enhanced analgesic properties of opioids are also observed in the setting of pain.

We thank Reviewer 1 for her/his valuable suggestion. However, the focus of the present paper is the analysis of opioid tolerance and side effects in phosphorylation-deficient mice. We strongly feel that we have now performed all relevant assays to effectively address this question. We agree with the Reviewer that opioid effects should also be analysed in the setting of pain in these animals once we have obtained animal ethics permission to perform

such studies. However, we feel that such extended studies are beyond the scope of the current study.

3. Finally, there are some procedures that need to be clarified and better justified. For example, opioid tolerance is induced following a 2-step procedure. First animals are injected with increasing concentrations of the opioid and then an osmotic minipump is inserted. The authors provide justification of the use of the minipump in order to maintain continuous opioid delivery but they don't provide justification for the intermittent administration of the drug prior to the insertion of the osmotic pump.

In response to the Reviewer 1, the following sentence was added on page 6 line 18: “Mice were given repeated cumulative doses of fentanyl or morphine to generate dose-response curves.” The intermittent administration of repeated cumulative doses is a very well established paradigm in the literature to generate dose-response curves from small groups of animals and to reduce the number of animals, which need to be implanted with osmotic minipumps. Dose-response curves generated before (day-1) and after (day 7) implantation of minipumps were then used to calculate the fold rightward shift of the dose-response curves, which is an accurate and reliable read-out for the development of tolerance. The pre and post cumulative dose-response curves are thus needed for a within-subject design in order to assess the influence of chronic opioid delivery on tolerance.

Reviewer 2 comments

This manuscript describes the effects of opioid agonists in knock-in mice where the C-terminus of the mu receptor (MOP) has had the serine/threonine residues mutated to alanines, and so cannot be phosphorylated in the C-terminus. The inference is that these mutant receptors cannot recruit β arrestins, and this has been shown in previous work in cell lines. The results of the present study clearly show that under these conditions, and compared to wild type mice, MOP in the mutant mice are less able to undergo agonist-induced desensitization over a short time frame of a few minutes. In addition the agonists fentanyl and morphine are more potent and longer lasting when tested for acute analgesia, and importantly other effects of these agonists (respiratory depression, constipation and locomotor activity) are not reduced in the absence of MOP C-terminus phosphorylation, and presumably β arrestin recruitment. Furthermore whilst the long term analgesic tolerance to fentanyl and to some extent morphine, is also removed or reduced, the withdrawal effects when naloxone is administered are not significantly altered. In general the quality of the experimental work is high and clear conclusions can be reached. The findings are of importance for a number of reasons Page 4 of 7 The findings are of importance for a number of reasons

-The opioid receptor and the associated drugs such as morphine and fentanyl are of very great importance in both medicine (analgesia etc) and society (drug abuse). -The molecular mechanism of opioid tolerance and its relationship to receptor phosphorylation is of great interest in the opioid and more general GPCR field

-For many years the opioid field has been influenced by previous findings implying that G proteins mediate MOP agonist analgesia, whilst β arrestins mediate the adverse effects such as respiratory depression. This idea has really dominated the opioid field for well over a decade, but the present results clearly overturn that idea. The work is thus very important and will have significant influence on the future direction of the opioid field.

1. Should not fentanyl and morphine have been used for the brain slice recording? Why was Met Enk used instead? At least some results with fentanyl and morphine should be included to show that these behave similarly to MetEnk.

Brain slice recordings have been performed with both morphine and fentanyl. However, these experiments are technically challenging for two reasons. First, the current induced by morphine is very small in the mouse so the measures are less reliable. Given that there is very little desensitization with morphine in the rat where the current is substantially larger, it is not

possible to obtain reliable measures of morphine-induced desensitization even in slices from wild type mice. Second, both morphine and fentanyl do not wash out of the slice in a reasonable time. In fact, in order to wash fentanyl from the bath and superfusion lines it requires cleaning with a dilute solution of DMSO (10 %). Thus, although these experiments would be very useful, it is not possible to include these compounds in experiments using brain slices.

2. Presumably it has been done but are brain GRK and β arrestin levels the same in the different mouse lines?

We have performed Western blot analyses of β -arrestin1 and GRK2 in brain. Our results show similar levels of these proteins in the different mouse lines and have been included Supplementary Fig. 2. We have added a comment on page 5 line 14.

3. It would have been useful to have some further brain biochemistry results to support the findings. For example, in the different mouse lines what happens to brain cAMP levels with naloxone challenge after chronic opioid? The behavioural results predict that the effects will be the same in all the mouse lines. Also it would be really good if it could be shown that in neurones from the mouse brains the mutant receptors do not recruit β arrestins. Is it possible that limited recruitment of arrestins could occur due to receptor activation alone, and independent of GRK-phosphorylation?

We agree with the Reviewer. A biochemical analysis of brain cAMP levels has not been performed. Consequently, we have deleted the statement on page 10, line 9: “suggesting that in the chronic presence of opioids, these mice may experience biochemical adaptations leading to dependence similar to those occurring in WT mice.“ Arrestin recruitment has been analysed in detail in our recent report (Ref. 13).

4. Do the mutant receptors still recruit GRKs? If so could this mediate effects independent of MOP phosphorylation? Could the mutant C-terminus of MOP interact with novel proteins?

Phosphorylation-deficient MOP receptors still recruit GRKs but to a much lesser extent than. This has been analysed in detail in our recent report in Science Signaling (Ref. 13).

5. I am not exactly sure what Figure 4 is meant to show, or what we are meant to conclude from it. The R2 for morphine are significant whilst those of fentanyl are not, but what are we meant to take from this? These sort of correlations with 4 points don't seem that useful.

Phosphorylation-deficient mice show enhanced analgesia due to enhanced G protein signalling in the absence of phosphorylation-dependent arrestin recruitment and desensitization. Figure 4 shows that under these conditions ED₅₀ values for respiratory depression and constipation are also shifted to the left, strongly suggesting that opioid side effects are also mediated by G protein-dependent pathways in the absence of β -arrestin signalling. This conclusion is presented on page 8 line 17: "Taken together, our findings suggest that the sustained G protein signalling observed in total phosphorylation-deficient MOP knockin mice leads to enhanced analgesia and to a proportional increase in opioid side effects that does not support a role for β -arrestin signalling in severity of respiratory depression or constipation." We think that this is one important finding of the present study, which should be presented in detail.

6. In the title the work "exacerbation" does not seem to be strongly supported by the results. It may be better to alter this to "maintenance of" or something similar. Figure 3 and Ext data Figure 6 show that fentanyl and morphine become more potent, but the extent of say the respiratory depression is not increased. If the authors want to retain "exacerbation" then they should explain clearly why this description is justified in the discussion.

Given the leftward shift in ED₅₀ values for both respiratory depression and constipation determined in phosphorylation-deficient mice compared to WT mice (compare Supplementary Table 1 and Figure 4), we feel it appropriate to use the term exacerbation of side effects. This is certainly true for respiration, for which there is a roughly 3-fold increase in potency for both fentanyl and morphine in 11S/T-A versus a roughly two-fold increase for analgesia.

7. Figure 3 – it would be good to see a time course of the respiratory depression in the different mouse lines and for the two agonists – are these the same? These could be included in the Ext data. I am also not clear why vehicle treatment is not used for comparison in parts a and c? In part c, the time course of the S375A mutant is strange. Any comment? Also check the X-axis label in c – is it correct?

In response to Reviewer 2, we have included in Supplementary Fig. 7a plethysmographic measurements of respiratory rates 15 min for fentanyl or 30 min for morphine after application of equally effective analgesic doses in each mouse line. We agree that in the time course of locomotor activity in the open field test in the S375A mutant (Figure 3a, fentanyl) there is a higher variability but no statistically significant difference between lines. The X-axis label in Figure c is correct. Measurements were performed over 2.5 hours.

8. P12 lines 4-10. Be specific in the studies being referred to here – which is the work of Bohn and colleagues, which has dominated the field for years.

In response to Reviewer 2, we have specifically added the respected studies being referred to each regulatory element. The sentence reads now: “Such additional regulatory elements may include GRK5^{11,12}, β -arrestin-2²³, and protein kinase C (PKC)³² and c-Jun N-terminal kinase 2 (JNK2)^{33,34}.”

9. P13 lines 16-20 – I was amused to see the RAVE idea being resurrected here – it is fine to leave as is, but it has been largely/totally discredited really.

The RAVE hypothesis has been cited here solely to emphasize the fact that up to now it was not possible to determine the exact contribution of either MOP desensitization or cellular adaptations to the development of tolerance. Phosphorylation-deficient mice are the first in vivo models, which allow this differentiation. We therefore feel it is appropriate to cite the RAVE hypothesis here.

10. Figure 5 – from visual inspection of the data, it seems to me that there is still significant tolerance to morphine in the mutant mice, which doesn't seem that different from the wild type when the shift is considered. Any comment on this?

We agree that tolerance still develops in 10S/T-A and 11S/T-A but it is significantly less pronounced compared to WT mice. (Please compare Table 1.)

11. P9 lines 7-8 – Some data should be included to show that the chronic drug treatment did not affect receptor levels. There is conflicting data on this in the literature.

In response to Reviewer 2, we have performed quantitative autoradiographic binding studies using [³H]DAMGO in brain sections from naïve transgenic mice and after chronic morphine or fentanyl treatment. Our results did not reveal any significant differences between genotypes or treatment groups. The additional results are depicted in Supplementary Fig. 1, Page 5, line 11 and page 9, line 17. We have added a comment on page 11 line 18 “The lack of change in MOP density across genotypes in naïve and chronic morphine and fentanyl treated mice indicates that any phenotypic alterations observed in these animals are not due to alterations in opioid receptor levels.”

12. Figure 5 – The global score for morphine in 11S/T-A is somewhat less than for the others – any reasons for this? Also it is not clear what we are meant to conclude from the weight loss data in general, e.g. the difference between morphine in 11S/T-A and wild type?

In the global score, we calculated jumping, grooming and wet-dog shakes, which are all withdrawal signs determined by visual inspection. The variability in this assay is inherently high and we believe it is a chance occurrence that combined (global) scores look a little (not significantly) lower than 10S/T-A or S375A. Visual inspection of Supplementary Figures bears this out, again with no differences in individual signs except for a reduction in grooming bouts for fentanyl (again not significant for global score). Weight loss was depicted separately because it is a typical withdrawal sign, which is not usually included in the global score. It is interesting that, if anything, weight loss is worse in the phosphorylation-deficient mutants. A more detailed analysis of factors influencing withdrawal-induced weight loss in the different mouse models will be part of future studies.

13. P2 line 18 – better to say at end of line “but that severe adverse effects are retained”?

In response to Reviewer 2, we have reworded this sentence. Page 2. Line 16 reads now: **Our results also predict that although G protein-biased opioid agonists may produce more effective analgesia and less tolerance, they are still likely to have severe adverse effects.** “

14. P3 line 12-14. Sentence grammatically incorrect.

Page 3 line 13-14 reads now: **“However, the individual contributions of MOP desensitization and cellular adaptation, and the molecular mechanisms initiating analgesic tolerance remain unresolved.”**

15. P6 line 4 – mention type of test used at beginning.

This sentence reads now: **„We then compared acute antinociceptive responses in S375A, 10S/T-A, 11S/T-A and WT mice in the hot-plate test.”**

16. P6 line 8 – are stats included for the heterozygous mice? Can’t see in the Ext data.

Stats were added to Supplementary Fig. 4.

17. P6 and elsewhere. The EC50 values depend on the cut-off for analgesia testing. Whilst the data presented is fine, this consideration should be mentioned somewhere. A statement such as “maximal possible analgesic effect” on P6 line 11 can be a little misleading.

We thank Reviewer 2 for her/his valuable suggestion. Page 7 line 2 reads now: **“The 50% effective dose (ED₅₀) values determined in the hot-plate test strongly depend on the temperature, which was set at 56°C, and cut-off for analgesia testing, which was set at 30 s.”**

18. P7 lines 5 and 8 – not sure constipation is normally regarded as a lethal effect with opioids?

In fact, opioid-induced constipation can cause serious clinical problems, which may lead to lethality due to induction of ileus and/or peritonitis.

19. P7 lines 21-22 – better to say that at low opioid drug concentrations they produced sig greater resp depression and constipation?

This sentence reads now, page 8, line 10: “Notably, we observed that 10S/T-A and 11S/T-A mice exhibited significantly greater respiratory depression and constipation than WT mice.”

20. P8 line 8 – again not sure it is right to refer to a “proportional increase in opioid side effects”

Please see point 5 for detailed response.

21. Fig5 – how were chronic doses decided upon/calculated?

In the many tolerance studies, twice daily injection of 10 mg/kg morphine is used to induce tolerance. Morphine has a duration of action of less than 4 hours in mice (Fig. 2d). To avoid extended drug-free intervals, we used implantation of osmotic minipumps with constant daily delivery of 17 mg/kg morphine. For fentanyl, we selected constant daily delivery of 2 mg/kg because of its considerably shorter duration of action (Fig. 2c).

22. P9 line 2 – “a significant shift in their” better

The sentence reads now: “Under identical conditions, S375A, 10S/T-A and 11S/T-A mice did not exhibit a significant shift in their sensitivity to fentanyl, suggesting that tolerance to fentanyl was abrogated in these animals (Table 1, Fig. 5b).”

23. P30 line 12. N=4 here but the data in panel d shows many more experiments (dots). Not clear why this is.

Panel C just shows representative recordings from 4 different mice. Consequently, N=4 was deleted from the description of panel C.

24. P31 line 13 – “equally effective doses for analgesia of...”

We agree and have corrected the sentence.

25. P32 line 14 – b,c rather than a,c I think

We agree and have corrected accordingly.

26. P32 line 20 and Fig 5 – does naloxone produce any scores if given to untreated mice, or for that matter, does vehicle produce any effects in treated mice?

In our hands, naloxone does not produce any score if given to untreated mice nor does vehicle produce any effects if given to treated mice.

Reviewer 3 comments

General comments:

This is a very interesting paper that examines mechanistic aspects of opioid receptor signaling with important implications for opioid analgesia, side effects and drug discovery. The approach is highly innovative; the topic is current. The authors are highly regarded experts in this field. The authors show that mutating 10 (S/T) phosphorylation sites at the C-terminus of the mu-opioid receptor (MOR) prolongs and enhances analgesia, reduces opioid tolerance but enhances adverse side effects. These results might significantly influence current trends in the design of opioid agonists (e.g. G-protein biased ligands). However, some important data are missing. Whereas desensitization is clearly abolished/diminished in S/Tmutants, no direct evidence of phosphorylation-deficiency, internalization or beta-arrestin recruitment is shown here. If the authors have such evidence, this must be included in the MS. The discrepant results about the role of S375 in this paper compared to previous publications (Reference No. 14) need to be discussed, as they raise concerns about reproducibility of the data. There are also some important issues with data presentation (were some of the data presented multiple times without stating so?), study design and statistical analysis.

Specific comments:

1. The abstract (p.2 line 9) (and many text passages throughout the MS) contains statements about the lack of arrestin recruitment in these mutants. However, no data on arrestin recruitment (or receptor internalization/phosphorylation) are shown here. Such data need to be included in the present MS. Instead, the authors refer to a paper in press which is not available to date.

Arrestin recruitment in phosphorylation-deficient MOP receptors has been analysed in detail in our recent report (Ref. 13), which has now appeared in Science Signaling.

2. In the introduction the authors make several statements concerning the relevance of amino acid residues for the phosphorylation/desensitization/internalization of opioid receptors. Some of these statements are too general because they do not specify the cells/tissues/animals that were used to generate these data in previous studies (e.g. cell lines, brain regions, rats, mice, humans etc.). This important information needs to be added. Similarly, the terms “low-efficacy agonist” (morphine) and “high-efficacy agonist” (fentanyl) (p. 4, line 3-4) are too general. Which type of efficacy (analgesia, side effects,

animal species, cell line, signaling pathway) do the authors refer to? Do they really mean “efficacy” or, rather, “potency”?

In response to Reviewer 3, we have specified in which cell lines, species and tissues previous studies have been performed. Page 3, line 20, page 3, line 23 and page 4, line 11 red text. We agree our use of terms low and high efficacy was not sufficiently explicit. Our usage pertained to efficacy for phosphorylation and β -arrestin recruitment as now stated as “...agonists having low efficacy for phosphorylation and β -arrestin recruitment...”. This is mentioned on page 4, lines 2 - 14.

3. Fig 2, Fig 5: Are these the same data/animals that are shown in Fig. 4? If so, this should be stated. The cumulative dose regimen needs to be described in more detail: the individual doses need to be stated; are the same animals injected multiple times? Is this procedure stressful for the animals?

In response to Reviewer 3, this part of the methods section was amended to provide more detail: “*Acute analgesia and tolerance paradigms.* For acute analgesia testing, dose-response curves were generated after subcutaneous administration of repeated cumulative doses of fentanyl or morphine. For fentanyl withdrawal latencies were measured 15 min after a first dose of 0.02 mg kg^{-1} ; at this time point, animals were injected with 0.03 mg kg^{-1} fentanyl for a cumulative dose of 0.05 mg kg^{-1} . Antinociception was assessed after 15 min, and mice were again injected with 0.05 mg kg^{-1} fentanyl to yield a cumulative dose of 0.1 mg kg^{-1} . Antinociception was again assessed after 15 min, and mice were again injected with 0.1 mg kg^{-1} fentanyl to yield a cumulative dose of 0.2 mg kg^{-1} . Antinociception was again assessed after 15 min, and mice were again injected with 0.1 mg kg^{-1} fentanyl to yield a final cumulative dose of 0.3 mg kg^{-1} . After which withdrawal latencies were again measured after 15 min. For morphine withdrawal latencies were measured 30 min after a first dose of 3.75 mg kg^{-1} . At this time point, animals were repeatedly injected with 3.75, 15 and 30 mg kg^{-1} morphine resulting final cumulative doses of 7.5, 22.5 and 52.2 mg kg^{-1} , respectively. Dose-response curves generated using repeated cumulative dosing regimen are shown in Figures 2a and b. These curves were used to calculate ED_{50} values for analgesia shown in Table 1 and Figure 4. To induce opioid tolerance, mice were implanted one day later with Alzet osmotic minipumps containing the same drug that was used in the cumulative dosing with fentanyl ($2 \text{ mg kg}^{-1} \text{ day}^{-1}$) or morphine ($17 \text{ mg kg}^{-1} \text{ day}^{-1}$) at a delivery rate of $0.5 \text{ } \mu\text{l/h}$. Thus, in the

tolerance paradigm dose-response curves from acute analgesia testing are depicted as day -1 in Figure 5 and served as reference for the development of tolerance. On day 7, mice were again treated using a repeated cumulative dosing regimen with fentanyl (0.05, 0.05, 0.1, 0.1 mg kg⁻¹ for 10S/T-A and 11S/T-A and 0.05, 0.15, 0.1, 0.1 mg kg⁻¹ for S375A and 0.5, 0.5, 1, 2 mg kg⁻¹ for WT) or morphine (7.5, 15, 15 mg kg⁻¹ for 10S/T-A and 11S/T-A and 22.5, 37.5, 60 mg kg⁻¹ for S375A and WT), and hot-plate response latencies were assessed at the same time points as on day -1. ED₅₀ values calculated from day -1 and day 7 dose-response curves were used to calculate the fold rightward shift given in Table 1 as an measure of tolerance.”

4. On p. 6, line 18-20 it is stated: “Interestingly, S375A mice display greater antinociception than WT after fentanyl (but only during the first 30 min) not but after morphine (Table 1, Fig. 2)”: The lack of enhancement of morphine analgesia is in contrast to results previously published in Grecksch et al. 2011 (Fig. 4 in that paper). The authors should address this reproducibility issue. A direct comparison of the doses/effects is also made difficult by the confusion about the employed doses in the present paper.

Reviewer 3 refers to our Ref 14, which is titled “*Analgesic tolerance to high-efficacy agonists but not to morphine is diminished in phosphorylation-deficient S375A mu-opioid receptor knock-in mice.*” In the present paper, we were able to fully reproduce the findings on tolerance given as Figures 5, 6, and 7 in Ref. 14. Regarding acute analgesia, findings on fentanyl in Figure 4 b right panel in Ref 14 were also fully reproduced, i.e. all three doses of fentanyl showed significantly enhanced analgesia. In Figure 4 b left panel, one out of three doses of morphine showed enhanced analgesia. In the present studies, we only observed enhanced analgesia in time course studies at 60 min (Figure 2d) but no significantly enhanced morphine analgesia in dose-response studies at 30 min (Figure 2b). We feel that this difference will not affect the conclusions drawn in the present paper nor the conclusions drawn in Ref. 14.

5. p. 7 Line 10-12: Whereas MOR internalization has been investigated in earlier papers, direct effects on arrestin recruitment are not shown in the cited papers (No. 5-9). The results presented in paper 13 are not available to date (paper in press?) making the appropriate assessment of the present MS impossible. Similar to the introduction, whenever data are compared between the present and previous manuscripts, the assays

and tissues (cell lines, species, brain regions etc.) used in the different publications need to be clearly stated in order to enable the reader to adequately assess comparability.

Arrestin recruitment in phosphorylation-deficient MOP receptors has been analysed in detail in our recent report (Ref. 13), which has now appeared in Science Signaling. Please compare our response to point 2.

6. p. 7 line 14: "...administered equi-analgesic doses of fentanyl or morphine ...". Are these doses really equally effective in analgesia? In Fig. 2 the S375A mutant seems different at 0.2 mg/kg compared to WT (same dose), and to 10S/T-A and 11S/T-A at 0.1 mg/kg.

We agree. Doses were not exactly equianalgesic. We therefore reworded this sentence to: "... we administered *nearly equianalgesic doses* ...".

7. p. 7 line 21-22: "... that 10S/T-A or 11S/T-A mice experienced significantly greater respiratory depression and constipation than WT mice." It seems that this sentence is not accurate for fentanyl, since the ED50s for constipation do not significantly differ from WT. Also, despite the clear differences in the ED50s (Ext. Table 1) for respiratory depression and constipation, the dose response graphs in Ext. fig 6 do not clearly reflect such differences. Perhaps the authors should opt for a different graphical representation which highlights better the ED50s shifts and accurately describe the way they calculated the ED50s from such dose response experiments.

This statement is accurate for fentanyl- and morphine-induced respiratory depression. For constipation only morphine was significant. Consequently, the word constipation was deleted.

8. p. 8 line 1-2 (and Fig 4): the "highly significant correlation" seems to be the case for morphine only and not for fentanyl. It would be interesting to discuss this discrepancy.

In response to Reviewer 3, we deleted fentanyl from this sentence. We feel that correlation observed for fentanyl did not reach statistical significance because there was a more heterogeneous distribution among the four genotypes after fentanyl treatment. Nevertheless, these findings strongly support our conclusion that in the absence of β -arrestin signalling opioid side effects are mediated by G protein-dependent pathways.

9. p. 9 line 5-6 (Fig. 5): “However, the rightward potency shift observed in 11S/T-A and 10S/T-A mice was significantly less pronounced than that observed in WT and..” A statistical test demonstrating this significance is missing.

This statement is supported by statistical analysis presented in Table 1.

10. p. 9 line 14 (Fig 5): “phosphorylation-deficient mice do not become tolerant,” This statement should be rephrased, as the authors claim elsewhere “a modest level of tolerance to morphine remains.”

In response to Reviewer 3, we reworded this sentence to: “As tolerance is strongly diminished in phosphorylation-deficient mice, ...”

11. p. 10 line 17-18: “... which stimulates MOP phosphorylation and β arrestin-2 recruitment with a similarly high efficacy as fentanyl“. References to this statement are missing.

Arrestin recruitment in phosphorylation-deficient MOP receptors has been analysed in detail in our recent report (Ref. 13), which has now appeared in Science Signaling.

12. p. 11 line 1 and Fig. 2b: The authors should discuss the discrepancy between this result and those presented in Grecksch et al. 2011

Reviewer 3 refers to our Ref 14, which is titled “*Analgesic tolerance to high-efficacy agonists but not to morphine is diminished in phosphorylation-deficient S375A mu-opioid receptor knock-in mice.*” In the present paper, we were able to fully reproduce the findings on tolerance given as Figures 5, 6, and 7 in Ref. 14. Regarding acute analgesia, findings on fentanyl in Figure 4 b right panel in Ref 14 were also fully reproduced, i.e. all three doses of fentanyl showed significantly enhanced analgesia. In Figure 4 b left panel, one out of three doses of morphine showed enhanced analgesia. In the present studies, we only observed enhanced analgesia in time course studies at 60 min (Figure 2d) but no significantly enhanced morphine analgesia in dose-response studies at 30 min (Figure 2b). We feel that this difference will not affect the conclusions drawn in the present paper nor the conclusions drawn in Ref. 14.

13. p. 12 line 2-4: "... at the same time fail to recruit β -arrestin proteins, these mutants can be viewed as completely G protein-biased MOP receptors that do not show improved side effect profiles". These statements are not supported by the data included in the present MS. If the authors have direct evidence of lack of arrestin/internalization/phosphorylation and G-protein bias in the cells/tissues from these knockin animals, such evidence should be included in the MS or the statement would need to be revised.

Arrestin recruitment in phosphorylation-deficient MOP receptors has been analysed in detail in our recent report (Ref. 13), which has now appeared in Science Signaling.

14. p. 13 line 11-12: "... enhanced weight loss, indicates that phosphorylation-deficient mice develop the same biochemical adaptations as WT mice in the absence of tolerance". Without cAMP measurements, this statement is speculation, considering the very general/nonspecific observation of weight loss. The enhancement of weight loss (especially in the 11S/T-A mutants) should be more carefully investigated, particularly since the treatment/testing paradigm applied here seems to be quite challenging for the animals. Moreover, the effect seems to be very small especially in the case of the 10S/T-A mutants. Therefore, the employed statistical tests should be double-checked.

We agree with the Reviewer. A biochemical analysis of brain cAMP levels has not been performed. Consequently, we have deleted the statement on page 10, line 9: "suggesting that in the chronic presence of opioids, these mice may experience biochemical adaptations leading to dependence similar to those occurring in WT mice." A more detailed analysis of factors influencing withdrawal-induced weight loss in the different mouse models will be part of future studies, but is beyond the scope of the current investigation."

15. p. 19, line 1: A description of how the data in Fig. 1d were generated (formulas of the ratios, normalization, which time points e.g. in the case of post/pre were compared etc.) would be helpful.

Going from left to right (1) "the decline from the peak was calculated by measuring the current at the end of a 10 min application of ME (30 μ M) divided by peak current induced shortly after (within 2 min) the application of ME (30 μ M)." This is a measure of the decrease in the maximum current that is induced by a saturating concentration of ME. (2) "The post/pre

measure is the amplitude of the current induced by ME (1 μ M) 5 min following the washout of ME 30 μ M divided by the initial current induced by ME (1 μ M).” This is a more sensitive assay to measure the decrease in sensitivity of ME following the ME (30 μ M, 10 min) desensitization period. (3) “The recovery from the desensitization induced by ME (30 μ M, 10 min) was measured at 5, 10 and 20 min following the washout of the high concentration of ME. The amplitude of the current induced by ME (1 μ M) was measured at each time point and normalized to the amplitude of the current induced by ME prior to the application of the high concentration of ME.” This measure shows that the desensitization at least partially recovers over a period of 20 min in slices from wild type and S375A animals – and remains the same in slices from the 10 and 11 ST/A animals. Thus experiments in slices from the phospho-deficient animals indicate an almost complete inhibition of acute desensitization.

16. p. 19, line 5-10: Were the animals used for the minipump implantation the same that were tested in Fig. 2 and 5? Were the chronic opioid administration experiments blinded?

In response to Reviewer 3, we provide more details in the methods section. Compare point 3. It has been made clear that acute analgesia was tested in animals, which received minipumps for the tolerance paradigm. Thus, data in Fig. 2 a and b are identical to Day -1 data in Fig. 5. Page 19, line 15 states: “The experimenter was blinded to treatment and/or genotype throughout the course of behavioural testing.”

17. p. 20, line 4-7: To improve the readability of the paper and the understanding of the results, the doses/injection regimen employed for these tests should be mentioned here. If these animals were then subjected to minipump implantation and to subsequent testing, this should be stated.

In response to Reviewer 3, we provide more details in the methods section. Compare point 3.

17. p. 20, line 9-11: there must be several typos in the indication of the doses employed.

We have checked and corrected this.

18. From Fig. 2b it seems that also 50 mg/kg was employed

We agree. The final cumulative dose was 52.5 mg/kg.

19. p. 22, line 21: Normal distribution of the data must be verified before performing parametric statistical analysis.

Page 23 lines 1-4: “Normal distribution of the data was verified before performing parametric statistical analysis. Wherever appropriate, data were analysed using one-way or two-way ANOVA, followed by Bonferroni’s or Dunnett’s *post hoc* tests or unpaired, two-tailed *t* test with significance set at $P < 0.05$.”

20. p. 24, line 4-6. This paper is not available to date (in press?). No data on arrestin recruitment are shown in the present MS. However, such data are essential for adequate evaluation of the present results, particularly when it comes to the patterns of recruitment of beta-arrestin by the activated phosphorylation-deficient MOP.

Arrestin recruitment in phosphorylation-deficient MOP receptors has been analysed in detail in our recent report (Ref. 13), which has now appeared in *Science Signaling*.

21. Fig. 1c: Which time point is represented by the last peak at the right side?

The time point of the last peak is 20 min.

In addition to the above changes, we have added Figure 6 as a schematic summary of our results.

Figure 6 Phosphorylation-deficient, G protein-biased MOP receptors enhance analgesia but worsen side effects. Carboxyl-terminal multi-site phosphorylation is the key step that drives acute MOP receptor desensitization and long-term tolerance. In the absence of MOP phosphorylation, arrestin recruitment is increasingly impaired yet phosphorylation-deficient MOP receptors can mediate opioid side effects such as respiratory depression, constipation and dependence.

We would like to take this opportunity to thank the Reviewers and the Editorial Board again for their valuable comments and criticisms, which helped us much to improve our paper. We hope that the revisions made will now permit publication of our manuscript in Nature Communications. Our manuscript contains original work and is not under consideration elsewhere. We appreciate the time spent by the Reviewers and the Editor, and we look forward to learn whether the manuscript is now acceptable for publication.

Sincerely yours,

Stefan Schulz, M.D.

Reviewers' comments:

Reviewer #1 (Remarks to the Author):

In the revised version of this manuscript the authors have provided some additional studies and clarifications. However, I feel that my concerns have not been properly addressed. These were related to the examination of the rewarding properties of these phosphorylation-mutant mice and whether the observed enhancement in opioid analgesia in the mutants could also be reproduced in pain model.

1) To address my first concern they conducted CPP studies for both fentanyl and morphine in WT and 11S/T-A mice. Data obtained are shown in Supplementary figure 8. However, I have some concerns and I'm not totally satisfied with the results that the authors are providing in this regard. For example, I feel that the graphical representation provided by the authors is misleading. On the Y-axis " Δ Time spent in drug-paired chamber" as it stands should refer to the time spent in morphine/fentanyl compartment during test-pre-conditioning to be accurate. However, the authors represent here the time in drug chamber – time in saline chamber. While this way of representing preference score virtually amplifies the drug-seeking effect (please prefer time in drug chamber post conditioning – pre-conditioning), the data presented in the revised version of the manuscript does not support the conclusion drawn by the authors page 8 line 6. A proper post-hoc analysis following the two-way ANOVA is mandatory to assess preference for the drug chamber. Considering the SEM presented in the figure, I highly doubt that any significance can be reached using proper statistical analysis. It is also unclear why the authors decided to run the post-conditioning test twice, 1 and 3 days after the last conditioning session, when it's quite clear from the data presented that there is no preference during the first post-conditioning test.

2) In their answer to my comment regarding the need to provide some additional data using a pain model, the authors mentioned that "the focus of the present paper is the analysis of opioid tolerance and side effects in phosphorylation-deficient mice". This seems in contradiction with the actual title of the paper : "Enhanced opioid analgesia and loss of tolerance but exacerbation of side effects in mice expressing phosphorylation-deficient, G protein-biased μ -receptors" and the first sentence from both their abstract or their introduction claiming the major use of opioids in pain relief. Thus, it seems really odd that the authors declined the proposed investigation on opioid analgesic properties in conditions of pain using their newly generated mice models. I would like to encourage the authors in pursuing such investigation.

3) Lastly, and as a general comment I feel that the study needs to be powered and strengthened to be able to challenge the actual dogma and the entire mu-opioid bias agonist field. While a significant amount of work has been conducted in this study, I also feel that, in its current state, the data presented in the manuscript does not provide strong evidence to fully challenge years of work on mu-opioid biased agonists.

Reviewer #2 (Remarks to the Author):

Overall this is an extremely interesting study with very important conclusions for the opioid field. The authors have answered all the points I raised in full, altering or adding to the text where appropriate.

Reviewer #3 (Remarks to the Author):

Authors' answers:

1. Arrestin recruitment in phosphorylation-deficient MOP receptors has been analysed in detail in our recent report (Ref. 13), which has now appeared in Science Signaling.

Re-revision:

There is a grammatical error in the abstract (line 1): "diminished". The abstract should explicitly state that data on arrestin recruitment and receptor phosphorylation are not shown in the present manuscript.

2. In response to Reviewer 3, we have specified in which cell lines, species and tissues previous studies have been performed. Page 3, line 20, page 3, line 23 and page 4, line 11 red text. We agree our use of terms low and high efficacy was not sufficiently explicit. Our usage pertained to efficacy for phosphorylation and β -arrestin recruitment as now stated as "...agonists having low efficacy for phosphorylation and β -arrestin recruitment...". This is mentioned on page 4, lines 2 - 14.

Re-revision:

There is a grammatical error on p. 3, line 5 ("drive"). The statements on p. 4, line 1 ("Agonists having low-efficacy for phosphorylation and β -arrestin recruitment such as morphine ...") and on p. 4, line 3 (... "high-efficacy opioids such as DAMGO and fentanyl ...") are still misleading because they suggest that effects on phosphorylation/arrestin recruitment are ligand-specific. However, these effects may well be specific for certain tissues, cells or species rather than particular ligands (see also Reviewer 2, point 1). The newly published paper (Ref # 13) suffers from similar problems because at least 3 different cell lines (HEK293, HEK293T, AtT20) were used there. This paper also mentions in the discussion "... the cellular effect of a particular ligand ... depends on the cellular context and that differences in expression levels of signaling and regulatory proteins will influence the downstream events of receptor activation in a particular cell type. For example, different expression levels of GRK2 may explain why morphine internalizes MOR in striatal neurons but not in other neurons ...". Therefore, the tissues/cell types used to demonstrate the various effects must be explicitly stated in the respective parts of the manuscript at hand. This information is still missing e.g. on p. 4, line 1 (after "phosphorylation"), on p. 4, line 3 (after "internalization"), on p. 4, line 6 (after "(GRK) 2 and 3"), on p. 4, line 7 (after "internalization"), on p. 4, line 10 (after "internalization"), on p. 4, line 14 (after "required") and on p. 4, line 16 (after "phosphorylation"). These considerations may affect the interpretation of data and must therefore be added to the discussion.

3. In response to Reviewer 3, this part of the methods section was amended to provide more detail: "Acute analgesia and tolerance paradigms. For acute analgesia testing, dose-response curves were generated after subcutaneous administration of repeated cumulative doses of fentanyl or morphine. For fentanyl withdrawal latencies were measured 15 min after a first dose of 0.02 mg kg⁻¹; at this time point, animals were injected with 0.03 mg kg⁻¹ fentanyl for a cumulative dose of 0.05 mg kg⁻¹. Antinociception was assessed after 15 min, and mice were again injected with 0.05 mg kg⁻¹ fentanyl to yield a cumulative dose of 0.1 mg kg⁻¹. Antinociception was again assessed after 15 min, and mice were again injected with 0.1 mg kg⁻¹ fentanyl to yield a cumulative dose of 0.2 mg kg⁻¹. Antinociception was again assessed after 15 min, and mice were again injected with 0.1 mg kg⁻¹ fentanyl to yield a final cumulative dose of 0.3 mg kg⁻¹. After which withdrawal latencies were again measured after 15 min. For morphine withdrawal latencies were measured 30 min after a first dose of 3.75 mg kg⁻¹. At this time point, animals were repeatedly injected with 3.75, 15 and 30 mg kg⁻¹ morphine resulting final cumulative doses of 7.5, 22.5 and 52.2 mg kg⁻¹, respectively. Dose-response curves generated using repeated cumulative dosing regimen are shown in Figures 2a and b. These curves were used to calculate ED₅₀ values for analgesia shown in Table 1 and Figure 4. To induce opioid tolerance, mice were implanted one day later with Alzet osmotic minipumps containing the same drug that was used in the cumulative dosing with fentanyl (2 mg kg⁻¹ day⁻¹) or morphine (17 mg kg⁻¹ day⁻¹) at a delivery rate of 0.5 μ l/h. Thus, in the

tolerance paradigm dose-response curves from acute analgesia testing are depicted as day -1 in Figure 5 and served as reference for the development of tolerance. On day 7, mice were again treated using a repeated cumulative dosing regimen with fentanyl (0.05, 0.05, 0.1, 0.1 mg kg⁻¹ for 10S/T-A and 11S/T-A and 0.05, 0.15, 0.1, 0.1 mg kg⁻¹ for S375A and 0.5, 0.5, 1, 2 mg kg⁻¹ for WT) or morphine (7.5, 15, 15 mg kg⁻¹ for 10S/T-A and 11S/T-A and 22.5, 37.5, 60 mg kg⁻¹ for S375A and WT), and hot-plate response latencies were assessed at the same time points as on day -1. ED50 values calculated from day -1 and day 7 dose-response curves were used to calculate the fold rightward shift given in Table 1 as an measure of tolerance.”

Re-revision:

These statements are sufficient but require language/grammatical corrections.

4. Reviewer 3 refers to our Ref 14, which is titled “Analgesic tolerance to high-efficacy agonists but not to morphine is diminished in phosphorylation-deficient S375A mu-opioid receptor knock-in mice.” In the present paper, we were able to fully reproduce the findings on tolerance given as Figures 5, 6, and 7 in Ref. 14. Regarding acute analgesia, findings on fentanyl in Figure 4 b right panel in Ref 14 were also fully reproduced, i.e. all three doses of fentanyl showed significantly enhanced analgesia. In Figure 4 b left panel, one out of three doses of morphine showed enhanced analgesia. In the present studies, we only observed enhanced analgesia in time course studies at 60 min (Figure 2d) but no significantly enhanced morphine analgesia in dose-response studies at 30 min (Figure 2b). We feel that this difference will not affect the conclusions drawn in the present paper nor the conclusions drawn in Ref. 14.

Re-revision:

These statements are sufficient and should be added to the discussion.

5. Arrestin recruitment in phosphorylation-deficient MOP receptors has been analysed in detail in our recent report (Ref. 13), which has now appeared in Science Signaling. Please compare our response to point 2.

Re-revision:

Please see comments on point 2.

6. We agree. Doses were not exactly equianalgesic. We therefore reworded this sentence to: “... we administered nearly equianalgesic doses ...”.

Re-revision:

This amendment is sufficient.

7. This statement is accurate for fentanyl- and morphine-induced respiratory depression. For constipation only morphine was significant. Consequently, the word constipation was deleted.

Re-revision:

This amendment is sufficient.

8. In response to Reviewer 3, we deleted fentanyl from this sentence. We feel that correlation observed for fentanyl did not reach statistical significance because there was a more heterogeneous distribution among the four genotypes after fentanyl treatment. Nevertheless, these findings strongly support our conclusion that in the absence of β -arrestin signalling opioid side effects are mediated by G protein-dependent pathways.

Re-revision:

This amendment is sufficient.

9. This statement is supported by statistical analysis presented in Table 1.

Re-revision:

This statement is sufficient.

10. In response to Reviewer 3, we reworded this sentence to: "As tolerance is strongly diminished in phosphorylation-deficient mice, ..."

Re-revision:

This amendment is sufficient.

11. Arrestin recruitment in phosphorylation-deficient MOP receptors has been analysed in detail in our recent report (Ref. 13), which has now appeared in Science Signaling.

Re-revision:

This amendment is sufficient.

12. Reviewer 3 refers to our Ref 14, which is titled "Analgesic tolerance to high-efficacy agonists but not to morphine is diminished in phosphorylation-deficient S375A mu-opioid receptor knock-in mice." In the present paper, we were able to fully reproduce the findings on tolerance given as Figures 5, 6, and 7 in Ref. 14. Regarding acute analgesia, findings on fentanyl in Figure 4 b right panel in Ref 14 were also fully reproduced, i.e. all three doses of fentanyl showed significantly enhanced analgesia. In Figure 4 b left panel, one out of three doses of morphine showed enhanced analgesia. In the present studies, we only observed enhanced analgesia in time course studies at 60 min (Figure 2d) but no significantly enhanced morphine analgesia in dose-response studies at 30 min (Figure 2b). We feel that this difference will not affect the conclusions drawn in the present paper nor the conclusions drawn in Ref. 14.

Re-revision:

These statements are sufficient and should be added to the discussion.

13. Arrestin recruitment in phosphorylation-deficient MOP receptors has been analysed in detail in our recent report (Ref. 13), which has now appeared in Science Signaling.

Re-revision:

Please see comments on point 2.

14. We agree with the Reviewer. A biochemical analysis of brain cAMP levels has not been performed. Consequently, we have deleted the statement on page 10, line 9: "suggesting that in the chronic presence of opioids, these mice may experience biochemical adaptations leading to dependence similar to those occurring in WT mice." A more detailed analysis of factors influencing withdrawal-induced weight loss in the different mouse models will be part of future studies, but is beyond the scope of the current investigation."

Re-revision:

This amendment is sufficient.

15. Going from left to right (1) "the decline from the peak was calculated by measuring the current at the end of a 10 min application of ME (30 μ M) divided by peak current induced shortly after (within 2 min) the application of ME (30 μ M)." This is a measure of the decrease in the maximum current that is induced by a saturating concentration of ME. (2) "The post/pre measure is the amplitude of the current induced by ME (1 μ M) 5 min following the washout of ME 30 μ M divided by the initial current induced by ME (1 μ M)." This is a more sensitive assay to measure the decrease in sensitivity of ME following the ME (30 μ M, 10 min) desensitization period. (3) "The recovery from the desensitization induced by ME (30 μ M, 10 min) was measured at 5, 10 and 20 min following the washout of the high concentration of ME. The amplitude of the current induced

by ME (1 μ M) was measured at each time point and normalized to the amplitude of the current induced by ME prior to the application of the high concentration of ME." This measure shows that the desensitization at least partially recovers over a period of 20 min in slices from wild type and S375A animals – and remains the same in slices from the 10 and 11 ST/A animals. Thus experiments in slices from the phospho-deficient animals indicate an almost complete inhibition of acute desensitization.

Re-revision:

These amendments are sufficient.

16. In response to Reviewer 3, we provide more details in the methods section. Compare point 3. It has been made clear that acute analgesia was tested in animals, which received minipumps for the tolerance paradigm. Thus, data in Fig. 2 a and b are identical to Day -1 data in Fig. 5. Page 19, line 15 states: "The experimenter was blinded to treatment and/or genotype throughout the course of behavioural testing."

Re-revision:

These amendments are sufficient.

17. In response to Reviewer 3, we provide more details in the methods section. Compare point 3.

Re-revision:

These amendments are sufficient.

18. We have checked and corrected this.

Re-revision:

These amendments are sufficient.

19. We agree. The final cumulative dose was 52.5 mg/kg.

Re-revision:

This amendment is sufficient.

20. Page 23 lines 1-4: "Normal distribution of the data was verified before performing parametric statistical analysis. Wherever appropriate, data were analysed using one-way or two-way ANOVA, followed by Bonferroni's or Dunnett's post hoc tests or unpaired, two-tailed t test with significance set at $P < 0.05$."

Re-revision:

These amendments are sufficient.

21. Arrestin recruitment in phosphorylation-deficient MOP receptors has been analysed in detail in our recent report (Ref. 13), which has now appeared in Science Signaling.

Re-revision:

Please see comments on point 2.

22. The time point of the last peak is 20 min.

Re-revision:

This amendment is sufficient.

Reviewer 1 comments

In the revised version of this manuscript the authors have provided some additional studies and clarifications. However, I feel that my concerns have not been properly addressed. These were related to the examination of the rewarding properties of these phosphorylation-mutant mice and whether the observed enhancement in opioid analgesia in the mutants could also be reproduced in pain model.

1. To address my first concern they conducted CPP studies for both fentanyl and morphine in WT and 11S/T-A mice. Data obtained are shown in Supplementary figure 8. However, I have some concerns and I'm not totally satisfied with the results that the authors are providing in this regard. For example, I feel that the graphical representation provided by the authors is misleading. On the Y-axis "Δ Time spent in drug-paired chamber" as it stands should refer to the time spent in morphine/fentanyl compartment during test-pre-conditioning to be accurate. However, the authors represent here the time in drug chamber – time in saline chamber. While this way of representing preference score virtually amplifies the drug-seeking effect (please prefer time in drug chamber post conditioning – pre-conditioning), the data presented in the revised version of the manuscript does not support the conclusion drawn by the authors page 8 line 6. A proper post-hoc analysis following the two-way ANOVA is mandatory to assess preference for the drug chamber. Considering the SEM presented in the figure, I highly doubt that any significance can be reached using proper statistical analysis. It is also unclear why the authors decided to run the post-conditioning test twice, 1 and 3 days after the last conditioning session, when it's quite clear from the data presented that there is no preference during the first post-conditioning test.

We thank Reviewer 1 for her/his comment to re-calculate the CPP data and to adopt a different graphical representation. The new graphs are shown in supplementary figure 8 (see below). On the top panel (a, b), total time spent in the drug-paired compartment during pre- and post-conditioning days is shown. On the bottom panel (c, d), the difference (Δ) in the time spent in the drug-paired between the post-conditioning and pre-conditioning days is depicted.

- I. In part a, b, we show the time in seconds which WT and 11S/T-A mice spend in the drug-paired compartment before (pre-conditioning) and after (post-conditioning) conditioning with fentanyl or morphine recorded over 15 min. We obtained significant

differences for both WT and 11S/T-A mice when comparing time spent in drug-paired chamber pre-conditioning vs post-conditioning separately for each genotype in a one-way ANOVA followed by Bonferroni's *post hoc* tests, * indicate statistically significant difference; fentanyl: $F_{(3, 44)} = 9.656$, $P < 0.0001$ and morphine: $F_{(3, 44)} = 9.529$, $P < 0.0001$. Thus, conditioned place preference clearly developed in both genotypes.

- II. In part c, d, we then calculated CPP as (time spent in the drug-paired side post-conditioning) – (time spent in the drug-paired side pre-conditioning). The Δ time-spent in the drug-paired compartment was statistically tested in a two-tailed *t* test by comparing WT vs 11S/T-A: fentanyl CPP ($P = 0.0807$) and morphine CPP ($P = 0.9438$). There was no significant difference in developing conditioned place preference between the genotypes.

In conclusion, both graphical representations suggest that conditioned place preference has been developed after fentanyl and morphine treatment without any differences between the genotypes.

We included the following sentence in the result section on page 9, line 4 from bottom: “Conditioned place preference was also assessed after fentanyl and morphine treatment. Both WT and 11S/T-A mice clearly developed conditioned place preference (Supplementary Fig. 8a, b). However, there was no significant difference between genotypes (Supplementary Fig. 8c, d).”

We have repeated the experiment two times for both morphine and fentanyl with 6 mice per genotype per experiment. For better representation of the data, we have shown in the new graph the result from each animal as individual points. We agree that there is no difference between the post-conditioning test on day 9 and day 11. For that reason it is sufficient to show only the results from the first post-conditioning test. All data shown are from post-conditioning day 9.

Supplementary Figure 8 Retained opioid seeking behaviour. a-d, On the pre-conditioning day, WT or 11S/T-A mice were assessed for time spent in one of the CPP compartments. Six days after the conditioning regime (0.1 mg kg⁻¹ fentanyl or 7.5 mg kg⁻¹ morphine on days 1, 3 and 5 and saline on days 2, 4 and 6) time spent in each CPP compartment was recorded again over 15 min (post-conditioning day). **a, b,** Pre- and post-conditioning time in drug-paired chamber of the conditioned place preference apparatus. Data are means ± s.e.m; * indicates statistically significant differences comparing pre- and post-conditioning time in the drug-paired chamber from each genotype (fentanyl: $F_{(3, 44)} = 9.656$, $P < 0.0001$ ($n = 12$) and morphine: $F_{(3, 44)} = 9.529$, $P < 0.0001$) ($n = 12$); one-way ANOVA with Bonferroni *post hoc* test. **c, d,** Data are shown as the difference (Δ) in time spent between the drug-paired compartment on the post-conditioning day and the pre-conditioning day. Data are means ± s.e.m; comparison of WT vs 11S/T-A in a two-tailed *t* test: fentanyl CPP ($P = 0.0807$) ($n = 12$) and morphine CPP ($P = 0.9438$) ($n = 12$).

We changed the supplementary method section on page 2 accordingly. The method section reads now: **“Conditioned place preference test (CPP).** The rewarding properties of opioids were measured in a 3-chambered CPP box (TSE Systems, Bad Homburg, Germany). Each of the three compartments was distinct. In the center compartment all walls and floor were light grey. One of the two “choice” compartment was white with a ribbed floor whereas the other compartment was striped black-and-white with a nub floor. The CPP procedure consisted of pre-conditioning (days 1-2, baseline on day 2), conditioning (days 3-8), and post-conditioning

(day 9). On days 1-2 and day 9, mice were allowed free access to all 3 chambers for 15 min. On day 2 (baseline), the amount of time mice spent on each side (left or right) during a 15-min testing period was recorded using a TSE Videomot system (TSE Systems, Bad Homburg, DE). Only mice with no significant place preference to one of the two compartments during pre-conditioning period underwent subsequent conditioning (unbiased) and were additionally drug conditioned to the less-preferred chamber^{42, 45}. During conditioning, the less-preferred compartment was paired to drug injection while the other compartment to saline injections. The conditioning phase (15 min per session with fentanyl and 30 min per session with morphine) was carried out daily over 6 consecutive days with alternate injections of drug or saline. WT and 11S/T-A mice were exposed to either fentanyl (0.1 mg kg⁻¹) or morphine (7.5 mg kg⁻¹) on days 3, 5 and 7 and saline on days 4, 6 and 8. On day 9, the mice were given free access to all 3 chambers for 15 min and time spent on each side (left or right) was recorded. We repeated the experiment two times for both fentanyl and morphine with 6 mice per genotype per experiment (N=12). CPP was calculated (Δ) as (time spent in the drug-paired side post-conditioning) – (time spent in the drug-paired side pre-conditioning)^{42, 45}. Data were analyzed using one-way ANOVA, followed by Bonferroni's *post hoc* tests or two-tailed *t* test with significance set at $P < 0.05$.”

2. In their answer to my comment regarding the need to provide some additional data using a pain model, the authors mentioned that “the focus of the present paper is the analysis of opioid tolerance and side effects in phosphorylation-deficient mice”. This seems in contradiction with the actual title of the paper: “Enhanced opioid analgesia and loss of tolerance but exacerbation of side effects in mice expressing phosphorylation-deficient, G protein-biased μ -receptors” and the first sentence from both their abstract or their introduction claiming the major use of opioids in pain relief. Thus, it seems really odd that the authors declined the proposed investigation on opioid analgesic properties in conditions of pain using their newly generated mice models. I would like to encourage the authors in pursuing such investigation.

Over the past decades, an array of different pain models has been developed. These can be differentiated into acute, chronic, inflammatory, neuropathic and tumor pain models. Acute pain is defined as a normal physiological response to external noxious stimuli. Acute phasic pain is typically measured using the hot plate test, tail flick test or Hargreaves thermal test. Acute tonic pain is typically measured using the formalin test, acetic acid writhing test or

capsaicin test. Acute inflammatory pain is typically measured using the complete Freund's adjuvant, carrageenan inflammation, zymosan paw edema or paw incision test. Neuropathic pain is typically measured using the chronic constriction, spared nerve injury or partial sciatic nerve ligation test. In response to Reviewer 1's comment, we have examined existing literature on prototypical G protein-biased μ agonists such as TRV130, PZM21 and SR17018 as well as classical reports on β -arrestin2-deficient mice, which is the gold standard genetic model for loss of opioid tolerance. In all of these studies, major conclusions were drawn based on results from acute thermal pain models such as hot plate or tail flick test. Consequently, we used the hot plate test in our study, which allows for an optimal comparability of our results with the results of the studies mentioned above as well as a wealth of earlier literature on opioid tolerance.

Given that the Reviewer did not specify her/his request for an additional pain model, we can only assume that Reviewer 1 is referring to acute tonic pain models. Nevertheless, it is generally accepted that in conditions of pain endogenous opioids will be increasingly released. In our mouse models, endogenous opioids released in conditions of pain will activate μ -opioid receptors with different desensitization behavior, i.e. rapidly desensitizing wild-type μ -receptors versus non-desensitizing phosphorylation-deficient mutants. It is conceivable that this fundamental difference will lead to different baseline values between different genotypes. An accurate assessment of opioid drug effects under these conditions will at least be very complicated and, we assert, is a very separate, very involved study that will not alter the conclusions of the studies presented here, regardless of outcome. Thus, we agree with Reviewers 2 and 3 that the conclusions drawn in the present paper regarding opioid analgesia and tolerance are very well justified based on our results from the hot plate test.

3. Lastly, and as a general comment I feel that the study needs to be powered and strengthened to be able to challenge the actual dogma and the entire mu-opioid bias agonist field. While a significant amount of work has been conducted in this study, I also feel that, in its current state, the data presented in the manuscript does not provide strong evidence to fully challenge years of work on mu-opioid biased agonists.

We were very surprised to see this new comment from Reviewer 1. In the initial review she/he correctly acknowledged that our study was aimed to provide necessary insight into the

mechanisms underlying μ -opioid receptor phosphorylation, desensitization and tolerance. Now, the same Reviewer feels that study needs to be powered and strengthened to fully challenge years of work on μ -opioid biased agonists. The authors feel that this is an unreasonable request for the following reasons:

- I. It is simply not possible to design a single study to fully challenge the μ -opioid biased agonist field and that was not the objective of our study.
- II. We do not see how additional – but unspecified - pain models suggested by Reviewer 1 could possibly power and strengthen the current paper in a way that would fully challenge years of work on μ -opioid biased agonists.
- III. Our study was by no means designed to challenge the μ -opioid bias agonist field and most importantly **we do NOT make that claim** in the manuscript.

The creation of phosphorylation-deficient mice enabled us for the first time to analyze opioid side effects mediated by G protein-biased μ -receptors. Our study was carefully conducted in four different laboratories located on three different continents to assure highest levels of reproducibility possible. We present a complete and novel data set on the characterization of three novel phosphorylation-deficient mouse lines. We are aware that some of the results are surprising if not disappointing for the μ -opioid biased agonist field. We concur with Reviewer 2 and 3 that these results should not be suppressed because our study as a whole provides very important conclusions for the opioid field.

We feel that our findings will contribute to the discussion on μ -opioid biased agonists and thus attract much attention. In fact, our study is not the first and will not be the last to potentially question current opinions on the μ -opioid biased agonist field. According to recent reports, the major concern about μ -opioid biased agonists is their retained or exacerbated abuse potential (Negus et al., Trends Pharmacol Sci. 2018, 39:916-919; Austin Zamarripa et al., Drug Alcohol Depend. 2018, 192:158-162). However, this is only part of the reason why the FDA recently rejected oliceridine's (TRV130) approval as pain killer. Other published studies found that prototypical μ -biased agonists can indeed induce respiratory depression in mice (Hill et al., Br J Pharmacol. 2018, 175:2653-2661). Moreover, a major weakness of the μ -opioid biased agonism hypothesis of reduced respiratory depression is that it is solely based on a single study on β -arrestin2-deficient mice (Raehal et al., J Pharmacol Exp Ther. 2005, 314:1195-201). In the opioid field, several independent groups have attempted to reproduce these findings but none of them succeeded. This observation will shortly be reported as a consensus paper. Finally, the real challenge for the entire μ -opioid biased agonist field will be to delineate the molecular pathways leading from arrestin recruitment to respiratory

depression and constipation *in vivo*. This work will involve creation of novel genetic models of μ -receptor neuron-specific β -arrestin1 and 2 knock out mice as well as creation of arrestin-biased μ -receptor mice etc. However, all currently of this work would be completely out of the focus of the present paper. In view of all available data on μ -opioid biased agonists, we are not persuaded to allocate additional resources into research on biased agonism as a strategy to develop safer opioids.

Reviewer 2 (Remarks to the Author):

Overall this is an extremely interesting study with very important conclusions for the opioid field. The authors have answered all the points I raised in full, altering or adding to the text where appropriate.

We thank Reviewer 2 for his/her valuable suggestions which helped us much to improve our manuscript.

Reviewer 3 comments

Authors' answers:

1. Arrestin recruitment in phosphorylation-deficient MOP receptors has been analysed in detail in our recent report (Ref. 13), which has now appeared in Science Signaling.

Re-revision: There is a grammatical error in the abstract (line 1): "diminished". The abstract should explicitly state that data on arrestin recruitment and receptor phosphorylation are not shown in the present manuscript.

We have corrected the grammatical error and changed the abstract accordingly. The abstract reads now: "Opioid analgesics are powerful pain relievers; however, over time, their pain control is diminished as analgesic tolerance develops, and life-threatening side effects limit their utility. The molecular mechanisms initiating tolerance remain unresolved to date. We have previously shown that rapid desensitization of the μ -opioid receptor (MOP) and its interaction with β -arrestin is controlled by hierarchical phosphorylation of multiple intracellular serine (S) and threonine (T) residues. Here, to assess the contribution ..."

2. In response to Reviewer 3, we have specified in which cell lines, species and tissues previous studies have been performed. Page 3, line 20, page 3, line 23 and page 4, line 11 red text. We agree our use of terms low and high efficacy was not sufficiently explicit. Our usage pertained to efficacy for phosphorylation and β -arrestin recruitment as now stated as "...agonists having low efficacy for phosphorylation and β -arrestin recruitment...". This is mentioned on page 4, lines 2 - 14.

Re-revision:

There is a grammatical error on p. 3, line 5 (“drive”). The statements on p. 4, line 1 (“Agonists having low-efficacy for phosphorylation and β -arrestin recruitment such as morphine ...”) and on p. 4, line 3 (... “high-efficacy opioids such as DAMGO and fentanyl ...”) are still misleading because they suggest that effects on phosphorylation/arrestin recruitment are ligand-specific. However, these effects may well be specific for certain tissues, cells or species rather than particular ligands (see also Reviewer 2, point 1). The newly published paper (Ref # 13) suffers from similar problems because at least 3 different cell lines (HEK293, HEK293T, AtT20) were used there. This paper also mentions in the discussion “... the cellular effect of a particular ligand ... depends on the cellular context and that differences in expression levels of signaling and regulatory proteins will influence the downstream events of receptor activation in a particular cell type. For example, different expression levels of GRK2 may explain why morphine internalizes MOR in striatal neurons but not in other neurons ...”. Therefore, the tissues/cell types used to demonstrate the various effects must be explicitly stated in the respective parts of the manuscript at hand. This information is still missing e.g. on p. 4, line 1 (after “phosphorylation”), on p. 4, line 3 (after “internalization”), on p. 4, line 6 (after “(GRK) 2 and 3”), on p. 4, line 7 (after “internalization”), on p. 4, line 10 (after “internalization”), on p. 4, line 14 (after “required”) and on p. 4, line 16 (after “phosphorylation”). These considerations may affect the interpretation of data and must therefore be added to the discussion.

We thank Reviewer 3 and agree. We have corrected the grammatical error on page 3 and changed the text on page 4 accordingly: “Within the ³⁷⁰TREHPSTANT³⁷⁹ cassette, S375 in the middle of this sequence is the primary site of phosphorylation *in vitro* in HEK293 and AtT20 cells^{6,8} as well as in mouse brain *in vivo*¹¹. Agonists having low-efficacy for phosphorylation and β -arrestin recruitment such as morphine induce a selective phosphorylation at S375 that does not facilitate receptor internalization either in HEK293 cells^{6,8} or mouse brain¹¹. After overexpression of GRK2 or GRK3 in cultured cells, morphine is able to promote multi-site phosphorylation and internalization of MOP^{8,12}. By contrast, high-efficacy opioids such as DAMGO and fentanyl not only induce phosphorylation of S375 but also drive higher-order phosphorylation on the flanking residues T370, T376, and T379 in a hierarchical phosphorylation cascade that specifically requires G protein-coupled receptor kinases (GRK) 2 and 3 in HEK293 cells¹² and in mouse brain¹¹. This multi-site

phosphorylation in turn promotes both β -arrestin recruitment and robust receptor internalization in HEK293 cells^{6,13}. In fact, mutation studies have revealed that mutations comprising all four S/T-residues within the ³⁷⁰TREHPSTANT³⁷⁹ cluster are necessary and sufficient to abolish β -arrestin recruitment and receptor internalization in HEK293 cells^{6,9,13}.”

In addition, we changed the discussion on page 11 accordingly: ”Although previous studies in transfected HEK293 and AtT20 cells have implicated phosphorylation by GRK2 and GRK3 and binding of the scaffolding proteins β -arrestin-1 and β -arrestin-2 in MOP desensitization^{6,7,9,13,26}, until now the contribution of carboxyl-terminal S/T phosphorylation to the physiological regulation of MOP has not been directly tested *in vivo*.”

3. In response to Reviewer 3, this part of the methods section was amended to provide more detail: “Acute analgesia and tolerance paradigms. For acute analgesia testing, dose-response curves were generated after subcutaneous administration of repeated cumulative doses of fentanyl or morphine. For fentanyl withdrawal latencies were measured 15 min after a first dose of 0.02 mg kg⁻¹; at this time point, animals were injected with 0.03 mg kg⁻¹ fentanyl for a cumulative dose of 0.05 mg kg⁻¹. Antinociception was assessed after 15 min, and mice were again injected with 0.05 mg kg⁻¹ fentanyl to yield a cumulative dose of 0.1 mg kg⁻¹. Antinociception was again assessed after 15 min, and mice were again injected with 0.1 mg kg⁻¹ fentanyl to yield a cumulative dose of 0.2 mg kg⁻¹. Antinociception was again assessed after 15 min, and mice were again injected with 0.1 mg kg⁻¹ fentanyl to yield a final cumulative dose of 0.3 mg kg⁻¹. After which withdrawal latencies were again measured after 15 min. For morphine withdrawal latencies were measured 30 min after a first dose of 3.75 mg kg⁻¹. At this time point, animals were repeatedly injected with 3.75, 15 and 30 mg kg⁻¹ morphine resulting final cumulative doses of 7.5, 22.5 and 52.2 mg kg⁻¹, respectively. Dose-response curves generated using repeated cumulative dosing regimen are shown in Figures 2a and b. These curves were used to calculate ED50 values for analgesia shown in Table 1 and Figure 4. To induce opioid tolerance, mice were implanted one day later with Alzet osmotic minipumps containing the same drug that was used in the cumulative dosing with fentanyl (2 mg kg⁻¹ day⁻¹) or morphine (17 mg kg⁻¹ day⁻¹) at a delivery rate of 0.5 μ l/h. Thus, in the tolerance paradigm dose-response curves from acute analgesia testing are depicted as day -1 in Figure 5 and served as reference for the development of tolerance. On day 7, mice were again

treated using a repeated cumulative dosing regimen with fentanyl (0.05, 0.05, 0.1, 0.1 mg kg⁻¹ for 10S/T-A and 11S/T-A and 0.05, 0.15, 0.1, 0.1 mg kg⁻¹ for S375A and 0.5, 0.5, 1, 2 mg kg⁻¹ for WT) or morphine (7.5, 15, 15 mg kg⁻¹ for 10S/T-A and 11S/T-A and 22.5, 37.5, 60 mg kg⁻¹ for S375A and WT), and hot-plate response latencies were assessed at the same time points as on day -1. ED₅₀ values calculated from day -1 and day 7 dose-response curves were used to calculate the fold rightward shift given in Table 1 as an measure of tolerance.”

Re-revision:

These statements are sufficient but require language/grammatical corrections.

In response to Reviewer 3 we have corrected the language and grammar. This part reads now: “***Acute analgesia and tolerance paradigms.*** For acute analgesia testing in the hot plate test, dose-response curves were generated after repeated subcutaneous administration of cumulative doses of fentanyl or morphine. Mice were injected at 15 min intervals with 0.02, 0.03, 0.05, 0.1 and 0.1 mg kg⁻¹ fentanyl to yield final cumulative doses of 0.02, 0.05, 0.1, 0.2 and 0.3 mg kg⁻¹ fentanyl, and latencies were measured 15 min after drug administration, immediately followed by additional drug except for the last dose. For morphine dose-responses, mice were injected at 30 min intervals with 3.75, 3.75, 15 and 30 mg kg⁻¹ morphine resulting in final cumulative doses of 3.75, 7.5, 22.5 and 52.5 mg kg⁻¹ morphine. Hot plate latencies were always measured 30 min after morphine administration, immediately followed by the next dose. Dose-response curves generated using repeated cumulative dosing regimen are shown in Figures 2a and b. These curves were used to calculate ED₅₀ values for analgesia shown in Table 1 and Figure 4. To induce opioid tolerance, mice were implanted one day later with Alzet osmotic minipumps containing the same drug that was used in the cumulative dosing. Osmotic minipumps delivered total daily doses of 2 mg kg⁻¹ fentanyl or 17 mg kg⁻¹ morphine at a rate of 0.5 µl/h. Thus, in the tolerance paradigm dose-response curves from acute analgesia testing are depicted as day -1 in Figure 5 and served as reference for the development of tolerance. On day 7, mice were again treated using a repeated cumulative dosing regimen with fentanyl (0.05, 0.05, 0.1, 0.1 mg kg⁻¹ for 10S/T-A and 11S/T-A and 0.05, 0.15, 0.1, 0.1 mg kg⁻¹ for S375A and 0.5, 0.5, 1, 2 mg kg⁻¹ for WT) or morphine (7.5, 15, 15 mg kg⁻¹ for 10S/T-A and 11S/T-A and 22.5, 37.5, 60 mg kg⁻¹ for S375A and WT), and hot plate response latencies were assessed at the same time points as on day -1. ED₅₀ values

calculated from day -1 and day 7 dose-response curves were used to calculate the fold rightward shift given in Table 1 as a measure of tolerance.”

4. Reviewer 3 refers to our Ref 14, which is titled ”Analgesic tolerance to high-efficacy agonists but not to morphine is diminished in phosphorylation-deficient S375A mu-opioid receptor knock-in mice.” In the present paper, we were able to fully reproduce the findings on tolerance given as Figures 5, 6, and 7 in Ref. 14. Regarding acute analgesia, findings on fentanyl in Figure 4 b right panel in Ref 14 were also fully reproduced, i.e. all three doses of fentanyl showed significantly enhanced analgesia. In Figure 4 b left panel, one out of three doses of morphine showed enhanced analgesia. In the present studies, we only observed enhanced analgesia in time course studies at 60 min (Figure 2d) but no significantly enhanced morphine analgesia in dose-response studies at 30 min (Figure 2b). We feel that this difference will not affect the conclusions drawn in the present paper not the conclusions drawn in Ref. 14.

Re-revision:

These statements are sufficient and should be added to the discussion.

We thank Reviewer 3 for her/his valuable suggestion. Page 7 reads now: “Significantly enhanced morphine analgesia in S375A mice was only observed in time course studies at 60 min (Figure 2d).”

5. Arrestin recruitment in phosphorylation-deficient MOP receptors has been analysed in detail in our recent report (Ref. 13), which has now appeared in Science Signaling. Please compare our response to point 2.

Re-revision:

Please see comments on point 2.

We have corrected the sentence on page 4. Please see point 2.

We hope that the revisions made will now permit publication of our manuscript in Nature Communications. We appreciate the time spent by the Reviewers and the Editor, and we look forward to learning whether the manuscript is now acceptable for publication.

REVIEWERS' COMMENTS:

Reviewer #1 (Remarks to the Author):

In this new version of the manuscript the authors satisfied most of my comments.

1) I would like to thank the authors for 1) the new data representation and statistics and 2) powering the effects on the place preference experiments.

2) I apologize for not mentioning the model of pain I was referring too. The acute phasic pain the authors are mentioning in their response represents only thermal sensitivity threshold. The stimulus applied in this case and according to the IASP definition cannot properly be referred as pain but rather to a discomfort, threshold sensitivity triggering a nocifensive behavior. The experiment I was mentioning was investigation on hyperalgesia/allodynia reversal in conditions of acute tonic/inflammatory or neuropathic pain. While I also understand that these hyperalgesia/allodynia measurements do not represent a direct measurement of current pain states but rather an increase in sensitivity to stimuli, they more closely relate to the conditions in which patients would be prescribed with opioid analgesics. I also agree with the authors that the results provided in the current manuscript are in line with the previous investigations on TRV130, PZM21 and SR17018 and mice lacking beta-arrestin2.

3) Lastly, I understand that it is not the intent, nor the goal of this manuscript to challenge the biased agonist field. I would like to mention again the significant amount of work provided in this manuscript, in generating new mice models and their proper characterization.